# Safe In-Context Reinforcement Learning

**Amir Moeini**[* 1]  **Minjae Kwon**[* 1]  **Alper Kamil Bozkurt**[2]  **Yuichi Motai**[2]  **Rohan Chandra**[1]  **Lu Feng**[1]
**Shangtong Zhang**[1]

## Abstract

In-context reinforcement learning (ICRL) is an emerging RL paradigm where an agent, after pretraining, can adapt to out-of-distribution test tasks without any parameter updates, instead relying on an expanding context of interaction history. While ICRL has shown impressive generalization, safety during this adaptation process remains unexplored, limiting its applicability in real-world deployments where test-time behavior is expected to be safe. In this work, we propose SCARED: Safe Contextual Adaptive Reinforcement via Exact-penalty Dual, the first method that promotes safe adaptation of ICRL under the constrained Markov decision process framework. During the parameter-update-free adaptation process, our agent not only maximizes the reward but also keeps the accumulated cost within a user-specified safety budget. We also demonstrate that the agent actively reacts to the safety budget; with a higher safety budget, the agent behaves more aggressively, and with a lower safety budget the agent behaves more conservatively. Across challenging benchmarks, SCARED consistently enables safe and robust in-context adaptation, outperforming existing ICRL and safe meta-RL baselines.

## 1. Introduction

Reinforcement learning (RL, Sutton & Barto (2018)) has achieved remarkable success in sequential decision-making, from game playing (Mnih et al., 2015) to robotic control (Levine et al., 2016). These achievements rely on deep neural networks whose parameters are updated through trial-and-error interaction with the environment. However, pa-

*Equal contribution [1]University of Virginia [2]Virginia Commonwealth University. Correspondence to:
Amir Moeini <amoeini@virginia.edu>,
Minjae Kwon <mjkwon@virginia.edu>.

*Proceedings of the 43rd International Conference on Machine Learning*, Seoul, South Korea. PMLR 306, 2026. Copyright 2026 by the author(s).

rameter updates at test time are not always feasible for adaptation: gradient computation may be unavailable, careful hyperparameter calibration can be required, and fine-tuning in meta-learning settings risks catastrophic forgetting of previously learned behaviors (Kirkpatrick et al., 2017).

Recently, in-context reinforcement learning (ICRL, Moeini et al. (2025)) has emerged as an RL paradigm that achieves this trial-and-error process without any neural network parameter updates (Duan et al., 2016; Xu et al., 2022; Lee et al., 2023; Laskin et al., 2023; Raparthy et al., 2024; Sinii et al., 2024; Huang et al., 2024; Krishnamurthy et al., 2024). Specifically, ICRL enables agents to adapt to new tasks at test time by conditioning on their interaction history, without updating network parameters. This allows a pretrained model to generalize to unseen environments with improving performance as more contextual experience accumulates.

Using ICRL in real-world deployments is especially desirable because, to enable adaptation, the infrastructure only needs to support the inference of the neural network. Despite the empirical success and the promising potential of ICRL (Laskin et al., 2023; Kirsch et al., 2023; Dai et al., 2024; Dong et al., 2024), the safety of ICRL has long been overlooked, which is a necessary condition for many possible real-world applications, such as embodied AI. In standard RL, agents are deployed on tasks they are trained for and expected to satisfy the safety constraints enforced during training. In contrast, in ICRL, agents explore and learn new tasks during test time. Therefore, safety constraints must be satisfied not just by the final learned behavior, but also during the adaptation process (Xu & Zhu, 2025). To our knowledge, no prior work has studied the safety of ICRL, which is the gap that this paper aims to close.

We model the safe ICRL framework as a constrained Markov decision process (CMDP, Altman (2021)), where agents receive not only a reward but also a cost signal after each action. We envision that safe ICRL agents should maximize the rewards while keeping the cost below a user-specified cost budget even when evaluated in out-of-distribution (OOD) tasks. In addition, they should actively react to the cost budget; e.g., a smaller cost budget should induce more conservative behavior for safety, while a larger cost budget should allow more aggressive behav-

ior for reward maximization. Meeting these desiderata is challenging; existing ICRL methods have no mechanism to incorporate cost signals, while safe meta-RL methods use past trial history only to fine-tune parameters, which does not capture the precise details of past experiences.

The key contribution of this work is to design the first safe ICRL agent that simultaneously fulfills the aforementioned desiderata, all without any parameter updates during the adaptation to new tasks. Specifically, we make the following contributions:

- **Safe ICRL Problem Formulation.** We formalize safe ICRL under the CMDP framework, where agents adapt to new tasks at test time without parameter updates while adjusting to safety budgets.

- **OOD Benchmarks for Safe In-Context Adaptation.** We introduce benchmark environments that evaluate safe exploration and extrapolative generalization beyond interpolation regimes under safety constraints.

- **SCARED: Safe Contextual Adaptive Reinforcement via Exact-Penalty Dual.** We propose SCARED, an online reinforcement pretraining method that satisfies CMDP safety constraints. We show SCARED outperforms the baselines, increasing cumulative reward and decreasing cost as context grows, while adapting to the safety budgets.

## 2. Background

### 2.1. Markov Decision Processes

We model the interaction with an environment for a given task as a finite-horizon MDP (Puterman, 1994). An MDP is defined by a state space $\mathcal{S}$, an action space $\mathcal{A}$, a reward function $r : \mathcal{S} \times \mathcal{A} \to \mathbb{R}$, a transition function $p : \mathcal{S} \times \mathcal{S} \times \mathcal{A} \to [0, 1]$, an initial distribution $p_0 : \mathcal{S} \to [0, 1]$, and a horizon length $T$. An agent interacts with the environment by following a policy $\pi : \mathcal{A} \times \mathcal{S} \to [0, 1]$. Specifically, after starting from an initial state $S_0 \sim p_0$, at any time step $t$, the agent takes an action $A_t \sim \pi(\cdot|S_t)$, observes a reward $R_{t+1} \doteq r(S_t, A_t)$, and proceeds to a next state $S_{t+1} \sim p(\cdot|S_t, A_t)$. This process continues until the time $T - 1$, when the agent takes a final action $A_{T-1}$ and receives the last reward $R_T$. We use $\tau \doteq (S_0, A_0, R_1, S_1, \ldots, S_{T-1}, A_{T-1}, R_T)$ to denote an episode, and use $k$ to index episodes and $t$ to index time steps in an episode. We write $S_t^k$, $A_t^k$, and $R_t^k$ to denote the state, action, and reward at time step $t$ in the $k$-th episode, respectively. For any episode $\tau$, we define its return as $G(\tau) \doteq \sum_{t=1}^{T} R_t$. The performance of a policy $\pi$ is then measured by the expected total rewards $J(\pi) \doteq \mathbb{E}_{\tau \sim \pi}[G(\tau)]$ obtained under $\pi$. The fundamental task in MDPs is to obtain an optimal policy $\pi$ that maximizes

$J(\pi)$, which typically requires a learning-based approach when a transition model is absent.

### 2.2. Reinforcement Learning

The goal of RL is to find an optimal policy for a given task by learning through interaction with the environment (Sutton & Barto, 2018). Classic approaches often construct tabular value functions that are used to explicitly map each state to an action, therefore are feasible only when the state and action spaces are discrete and small. Recent RL techniques, combined with modern deep learning methods can handle large or continuous state–action spaces via function approximation with neural networks parameterized by weights $\theta$ (Wang et al., 2024). Starting from an initial policy $\pi_{\theta_0}$, deep RL methods typically update the parameters using observed transitions $(S_t, A_t, S_{t+1}, R_{t+1})$ to obtain $\pi_{\theta_{t+1}}$. Performance can improve through these updates, which aim to reduce prediction errors via backpropagation through a loss function. These techniques have been successful at learning high-performing policies for a particular task; however, their performance can degrade substantially across OOD tasks and environments. In such settings, policies are often re-learned or fine-tuned, which limits the large-scale deployment of a single trained policy.

### 2.3. In-Context Reinforcement Learning

Recently, ICRL has emerged as a paradigm that allows RL agents to adapt to new tasks at test time without requiring any parameter updates after pretraining. In ICRL, the parameters $\theta_*$ are first learned with a pretraining procedure on multiple MDPs. The resulting policy $\pi_{\theta_*}$ parameterized by $\theta_*$ is then deployed to adapt to new tasks by taking context as additional input. We denote the context at time $t$ within the $k$-th episode as $H_t^k$ and we write the policy as $\pi(\cdot|S_t^k, H_t^k)$ to make this dependence explicit. A simple choice is to use the history of transitions as context (see Moeini et al. (2025) for alternative approaches); i.e., $H_t^k \doteq (\tau_1, \ldots, \tau_{k-1}, \tau_k^t)$ where $\tau_k^t \doteq (S_0^k, A_0^k, R_1^k, S_1^k, \ldots S_{t-1}^k, A_{t-1}^k, R_t^k)$ is the observed prefix of the current episode. Adaptation to a new task in ICRL occurs as the performance of $\pi_{\theta_*}(\cdot|S_t^k, H_t^k)$ improves with growing context while $\theta_*$ remains fixed. This generalization capability is hypothesized to be a consequence of the neural network implicitly implementing an RL algorithm in its forward pass, processing the context at inference time (Laskin et al., 2023; Kirsch et al., 2023). As the context grows, this inference-time algorithm gains access to more information, improving policy performance. This hypothesis is also theoretically supported by Lin et al. (2024); Wang et al. (2025a;b); Xie et al. (2026b;a). Despite the remarkable generalization of ICRL to OOD tasks, the safety of such generalization has been overlooked.

We now discuss the pretraining methods for ICRL, which

can typically be divided into supervised pretraining (Laskin et al., 2023; Lin et al., 2024) and reinforcement pretraining (Duan et al., 2016; Wang et al., 2017; Grigsby et al., 2024a;b; Wang et al., 2025a;b). Supervised pretraining employs behavior cloning with the goal of imitating an algorithm rather than a policy, and it is usually done in an offline manner. Specifically, by running an existing RL algorithm on an MDP, we can collect a sequence of episodes, denoted as $\Xi \doteq (\tau_1, \tau_2, \ldots, \tau_K)$. We call $\Xi$ a trajectory. The episode return $G(\tau_k)$ is expected to increase as $k$ increases; therefore, the trajectory $\Xi$ can be used to demonstrate the learning progression of the RL algorithm on the MDP. In supervised pretraining, multiple trajectories are collected by running multiple existing RL algorithms on multiple MDPs, yielding a dataset $\{\Xi_i\}$. The policy $\pi_\theta$ is then trained with an imitation learning loss. In other words, for a state $S_t^k$ from a trajectory $\Xi_i$ in the dataset, the loss for updating $\theta$ is

$$-\log \pi_\theta(A_t^k | S_t^k, H_t^k). \quad (1)$$

By optimizing $\theta$ to imitate the behavior of RL algorithms demonstrated in the dataset, the policy $\pi_\theta$ is expected to embed some RL-like mechanism into its forward pass, which is known as algorithm distillation (Laskin et al., 2023). Instead of distilling existing RL algorithms, reinforcement pretraining optimizes the network to maximize the return on many different MDPs, guiding it to discover its own RL algorithm by providing the interaction history. Reinforcement pretraining is typically done in an online manner. At time step $t$ within the $k$-th episode, the loss is

$$Loss_{RL}(\pi_\theta(\cdot | S_t^k, H_t^k)), \quad (2)$$

where $Loss_{RL}$ can be any standard RL loss for standard online RL algorithms, e.g., Grigsby et al. (2024a) use a variant of DDPG (Lillicrap et al., 2016), Elawady et al. (2024) use a variant of PPO (Schulman et al., 2017), and Cook et al. (2024) use Muesli (Hessel et al., 2021). Since the context is typically a long sequence, Transformers (Vaswani et al., 2017) or state space models (Gu & Dao, 2024) are usually used to parameterize the policy (Laskin et al., 2023; Lu et al., 2023). The long sequence context is one of the main driving forces for the remarkable generalization capability of ICRL.

## 3. Problem Formulation: Safe ICRL

We study the safety of ICRL using the CMDP framework (Altman, 2021). In addition to the reward function $r$, a CMDP involves a cost function $c : \mathcal{S} \times \mathcal{A} \rightarrow \mathbb{R}$ with an associated user-given budget $\delta$. At each time step $t$, after taking an action $A_t$ at a state $S_t$, the agent additionally receives a cost $C_{t+1} \doteq c(S_t, A_t)$. An episode in a CMDP therefore becomes

$$\tau = (S_0, A_0, R_1, C_1, S_1, \ldots, S_{T-1}, A_{T-1}, R_T, C_T). \quad (3)$$

We denote the total cost of an episode $\tau$ by $G_c(\tau) \doteq \sum_{t=1}^{T} C_t$, which should, in expectation, remain below the budget $\delta$.

The objective of ICRL is to pretrain a set of parameters $\theta_*$ such that the resulting policy $\pi_{\theta_*}$ maximizes the sum of expected returns for a new task at test time. This is usually achieved by the return $G(\tau_k)$ of an episode $\tau_k$ being increased with $k$, as the policy learns more about the environment. In safe ICRL, we additionally expect the episode cost $G_c(\tau_k)$ to decrease as $k$ increases, ideally remaining below a user-specified cost budget $\delta$.

We formalize this safe ICRL objective as follows:

$$\max_\theta \mathbb{E}_{\pi_\theta}[\textstyle\sum_{k=1}^{K} G(\tau_k)] \text{ s.t. } \forall k, \mathbb{E}_{\pi_\theta}[G_c(\tau_k)] \leq \delta. \quad (4)$$

Here, $K$ is the number of total test episodes, and $\tau_k$ denotes the $k$-th test episode obtained by executing the policy $\pi_\theta$ on a novel CMDP that is not accessible during pretraining. Throughout test time, the CMDP and the policy parameters $\theta$ are fixed; therefore, the policy $\pi_\theta(\cdot | S_t^k, H_t^k)$ can adapt to the CMDP only by utilizing the context, i.e., the history of test transitions. Although we formulate our problem using CMDPs, our formulation can be used for partially observable systems without loss of generality.

## 4. SCARED: Safe Contextual Adaptive Reinforcement Pretraining

We now introduce SCARED, our online reinforcement pretraining approach for the safe ICRL problem. SCARED enables safe in-context adaptation to unseen CMDPs by regulating episode-level costs within a given budget. We establish a pretraining procedure that minimizes the loss (2) across a range of CMDPs, extending the generalization capability from reward maximization to also respecting the safety budget, solely based on context without parameter updates at test time. Specifically, we design an algorithm to solve (4) by converting the primal problem to its dual form[1]:

$$\min_{\boldsymbol{\lambda} \succeq 0} \max_\pi L(\pi, \boldsymbol{\lambda}) =$$

$$\min_{\boldsymbol{\lambda} \succeq 0} \max_\pi \left[ \mathbb{E}_\pi \left[ \sum_{k=1}^{K} G(\tau_k) \right] - \sum_{k=1}^{K} \lambda_k (\mathbb{E}_\pi[G_c(\tau_k)] - \delta) \right]. \quad (5)$$

where $\boldsymbol{\lambda} \succeq 0$ denotes componentwise nonnegativity of the Lagrangian multipliers.

The maximization over $\pi$ ranges over contextual policies that condition not only on the interaction history but also on the total cost expected over the remainder of the episode. We refer to this total future cost as cost-to-go (CTG) and

---

[1]We omit $\theta$ and write $\max_\pi f(\pi)$ instead of $\max_\theta f(\pi_\theta)$ for brevity.

define it as $G_{c,t}(\tau) \doteq \sum_{i=t+1}^{T} C_i$. At the start of each episode, CTG is set to the budget $\delta$, i.e., $G_{c,0} = \delta$. Policies then sample actions conditioned on CTG as $A_t^k \sim \pi_\theta(\cdot \mid S_t^k, H_t^k, G_{c,t}(\tau_k))$.

In safe ICRL, applying a separate constraint and Lagrange multiplier to each evaluation episode $\tau_k$ is undesirable as this requires fixing the number of episodes at the start of pre-training and leads to uneven update frequencies across multipliers. For example, if we expect $K$ evaluation episodes, we must initialize at least $K$ multipliers. If pretraining rollouts contain fewer episodes than the number of multipliers, later multipliers are updated less frequently, which destabilizes the optimization (see Figure 5.(c)). Furthermore, the constraints should be symmetric since they are associated with a single cost function but simply correspond to different episodes. These motivate the use of a single multiplier for all episodes. However, using a single multiplier will over-penalize the policy on episodes that already satisfy the constraint. To prevent this, we propose a modified Lagrangian function and an iterative optimization scheme. We show this approach is empirically more stable while preserving the desirable optimization properties.

Let $g_k(\pi) := \mathbb{E}_\pi[G_c(\tau_k)] - \delta$. We consider the following surrogate objective for the policy

$$L_\Sigma(\pi, \lambda) = \mathbb{E}_\pi\Big[\sum_{k=1}^{K} G(\tau_k)\Big] - \lambda \sum_{k=1}^{K} [g_k(\pi)]_+$$

where $[x]_+ \doteq \max\{x, 0\}$ and perform the following iterative updates:

$$\begin{aligned} \pi_{t+1} &\in \operatorname{argmax}_\pi L_\Sigma(\pi, \lambda_t) \\ \lambda_{t+1} &= [\lambda_t + \eta \max_k g_k(\pi_{t+1})]_+ \quad \eta > 0. \end{aligned} \quad (6)$$

We analyze the updates in (6) under the following regularity assumptions.

**Assumption 1.** *The expected return $\mathbb{E}_\pi\Big[\sum_{k=1}^{K} G(\tau_k)\Big]$ and expected cost $\mathbb{E}_\pi[G_c(\tau_k)]$ are bounded and continuous in $\pi$. The constrained problem* (4) *admits an optimal feasible policy $\pi^\star$.*

**Assumption 2.** *[Paternain (2018)] Problem* (4) *satisfies conditions ensuring zero duality gap with* (5) *and there exist optimal Lagrange multipliers $\boldsymbol{\lambda}^\star \succeq 0$ such that*

$$\max_{\substack{\pi \\ \text{s.t. } \forall k, \mathbb{E}_\pi[G_c(\tau_k)] \leq \delta}} \mathbb{E}_\pi\Big[\sum_{k=1}^{K} G(\tau_k)\Big] = \min_{\boldsymbol{\lambda} \succeq 0} \max_\pi L(\pi, \boldsymbol{\lambda})$$

The following theorem shows that the set of the fixed points of (6) and the set of the optimizers of (4) are equal, meaning that the objective of our approach aligns with the safe ICRL problem statement.

**Theorem 1.** *[Proof in Appendix B.1] We say a pair $(\bar{\pi}, \bar{\lambda})$ is a fixed point of* (6) *if $\bar{\pi} \in \arg\max_\pi L_\Sigma(\pi, \bar{\lambda})$, and for all sufficiently small $\eta > 0$, $\bar{\lambda} = [\bar{\lambda} + \eta \max_k g_k(\bar{\pi})]_+$. Let Assumptions 1 - 2 hold. Then every primal-optimal policy $\pi^*$ admits a corresponding fixed point $(\pi^*, \bar{\lambda})$ for any multiplier $\bar{\lambda} \geq \|\boldsymbol{\lambda}^\star\|_\infty$, where $\boldsymbol{\lambda}^\star$ is an optimal dual solution. Conversely, every fixed point $(\bar{\pi}, \bar{\lambda})$ of* (6) *is feasible and primal-optimal for* (4)*.*

Our update (6) resembles an exact penalty convex optimization technique when $\lambda \geq \|\boldsymbol{\lambda}^\star\|_\infty$ (Nocedal & Wright, 1999), hence the naming. We implement it with an actor-critic approach following Grigsby et al. (2024a); see Algorithm 1 for pseudocode. Separate critics estimate the reward Q-function $Q_{\theta_v}$ and cost Q-function $Q_{\theta_c}^c$, trained with TD targets from target networks. The actor maximizes $Q_{\theta_v}$ while penalizing $Q_{\theta_c}^c$ for episodes exceeding the budget.

## 5. Experiments

We now empirically investigate the proposed safe ICRL algorithms, aiming to answer the following research questions:

(i) Does SCARED generalize to both out-of-distribution tasks and unseen in-distribution (ID) tasks in safety-constrained environments?

(ii) How effectively does SCARED achieve flexible reward-cost tradeoffs under varying safety budgets at test time, without parameter updates?

### 5.1. Baselines

We consider two baseline classes: algorithm distillation for safe RL, which encodes safety into the policy's forward pass without test-time parameter updates, and safe meta-RL methods, which adapt via parameter updates at evaluation but do not use in-context interaction histories for action selection.

**Algorithm Distillation for Safe RL.** We adopt algorithm distillation for safe RL (safe AD) as a strong baseline for safe ICRL. Following algorithm distillation (Laskin et al., 2023), we collect a dataset $\{\Xi_i\}$ by executing existing safe RL algorithms (e.g., (Tessler et al., 2019; Ray et al., 2019)) across safety-constrained environments (e.g., (Ji et al., 2023; Gu et al., 2025)). As a result, each trajectory $\Xi_i$ consists of multiple episodes and each episode contains both the reward and the cost (cf. (3)).

To ensure a fair and competitive baseline, the policy is conditioned on the full in-context interaction history $H_t^k$, which includes per-step reward and cost signals, as well as on CTG and return-to-go (RTG), a target specifying the desired future reward that steers the policy toward a desired reward level (Liu et al., 2023). The raw reward and cost information in $H_t^k$ enables algorithm dis-

tillation (Laskin et al., 2023), allowing the model to imitate adaptation behavior present in the offline trajectories. In contrast, RTG and CTG serve a distinct role as conditioning variables that bias action selection toward desired reward-cost tradeoffs at test time. Formally, given a trajectory $\Xi = (\tau_1, \ldots, \tau_K)$, the safe AD objective at state $S_t^k$ is $-\log \pi_\theta \left( A_t^k \mid S_t^k, H_t^k, G_t(\tau_k), G_{c,t}(\tau_k) \right)$. During pretraining, RTG and CTG are available since episodes are complete. At test time, they are replaced by user-specified targets, allowing control over the reward-cost tradeoffs without parameter updates. Safe AD relies on a large replay buffer of logged learning histories, which must be subsampled due to context-length constraints and labeled with RTG to distinguish between early, intermediate, and expert phases of training (Dai et al., 2024). We also evaluate a noise-based variant of Safe AD (Zisman et al., 2024); however, training on optimal trajectories with noise fails to provide the behavioral diversity required for OOD adaptation, highlighting the learning mechanism in our Safe AD baseline for reliable in-context safety adaptation (Appendix D).

**Safe Meta RL.** We compare our approach against gradient-based safe meta RL methods, specifically MAML with penalty (Finn et al., 2017), and SafeMeta (Xu & Zhu, 2025). MAML with penalty optimizes a model-agnostic initialization via cost-weighted penalties that can be adapted to new tasks with a few gradient steps. SafeMeta combines a closed-form, one-step safe policy adaptation method with a Hessian-free meta-training algorithm to guarantee anytime safety. By design, these methods adapt through per-task parameter updates rather than conditioning on cross-episode interaction histories.

### 5.2. Experimental Setup

**Safe ICRL Benchmarks.** We evaluate our approach on three constrained environments, categorized into two complementary generalization regimes: (i) structural OOD tasks and (ii) unseen ID tasks. First, we use SafeDarkRoom and SafeDarkMujoco (Point and Car) to evaluate adaptation to OOD tasks with shifted goal and obstacle configurations. Second, we evaluate on SafeVelocity (HalfCheetah and Ant), representing the unseen ID task regime commonly used in meta-RL benchmarks. We discuss the difficulty gap between these two categories in Section 5.3.

The first two environments are modified from Dark-Room (Laskin et al., 2023) and SafetyGym benchmark (Ji et al., 2023) respectively. SafeDarkRoom is a grid-world setting where the agent can only perceive its own position and lacks visibility of the goal or obstacle locations. Rewards are sparse, with a positive reward obtained upon reaching the goal in the map. Costs arise from multiple obstacles scattered across the environment. For instance, 25 obstacles in a $9 \times 9$ grid map, which incur a cost of 1 each time the agent

steps on one of the obstacles. To succeed, the agent must learn obstacle positions based on encountered cost signals, introducing additional complexity compared to goal-only environments due to the presence of multiple hazards.

SafeDarkMujoco operates in continuous space with MuJoCo simulation (Todorov et al., 2012), with a setup analogous to SafeDarkRoom. The robot senses its internal physical states, such as velocity, acceleration, position, and rotation angle, while range-based sensing (e.g., lidar) is unavailable, preventing direct detection of obstacles and the goal. Consequently, the agent must develop exploratory behaviors to identify obstacles and goals, akin to the SafeDarkRoom design, while navigating to the goal and minimizing collisions by learning from previous collisions in the context. SafeVelocity follows the design of Ji et al. (2023); however, we vary the target velocities and hold out a set of unseen velocities for evaluation. Such velocity-conditioned setups are often considered within meta-RL frameworks (Finn et al., 2017), where different target velocities are sampled from a fixed range (e.g., $[0, 1]$). We thus consider SafeVelocity tasks as unseen ID tasks, which are more straightforward than OOD generalization.

**Training Setup.** For OOD task environments, all algorithms are trained on source tasks with center-oriented obstacles and goals. For unseen ID task environments, we hold out a set of target velocities multipliers uniformly sampled from $[0.5, 1.0]$ in increments of 0.05 during training and evaluating performance on these held-out velocities at test time.

Safe AD is trained on trajectories generated using PPO-Lagrangian (Ray et al., 2019). For SCARED, we perform online reinforcement pretraining for 30,000 steps on SafeDarkRoom and 10,000 steps on SafeDarkMujoco using Algorithm 1, and we train for 1,000 steps on the SafeVelocity environments (HalfCheetah and Ant). Safe meta-RL baselines are trained for up to 15,000 steps on each environment, with most converging earlier, using their respective update rules. For safe AD and safe meta-RL baselines, we perform hyperparameter optimization to ensure fair comparison across methods. Additional implementation details and used hyperparameters are provided in Appendix C.

**Evaluation Setup.** To answer the aforementioned research questions, we use a consistent setup for CTG and RTG during evaluation. For SCARED, we evaluate robustness to safety budgets by conditioning on CTG values sampled uniformly from $[1, 10]$ in SafeDarkRoom, $[10, 50]$ in SafeDarkMujoco, and $[0, 5]$ in SafeVelocity, reporting performance averaged over 100 distinct CTG targets. This validates that pretraining with diverse CTG targets enables budget-conditioned generalization to OOD tasks. The same intervals and sample counts are used for the cost budgets of SafeMeta and MAML with penalty. For Safe AD, we condition the policy on paired target values. Specifically,

we initialize RTG at $0.1$ and CTG at $0.0$, and then adjust RTG according to $\text{RTG} = \max\left\{0.1, \frac{\text{CTG}}{\text{CTG}_{\max}}\right\}$ as CTG increases from 0 to $\text{CTG}_{\max}$, where $\text{CTG}_{\max} = 10$ for SafeDarkRoom, 50 for SafeDarkMujoco, and 5 for SafetyVelocity. Performance is reported as an average over 100 paired RTG-CTG targets.

## 5.3. Measuring OOD generalization

Goal discovery-oriented environments such as DarkRoom are widely used in previous ICRL works (Laskin et al., 2023; Zisman et al., 2024; Son et al., 2025). In those works, the goals are randomly spawned over the map. Each goal corresponds to a new task. By ensuring the goals used in pretraining do not overlap with the goals used in testing, they ensure the test task is unseen during pretraining and can thus evaluate the generalization capability of their ICRL agents. However, such generalization can be achieved by interpolation (Kirk et al., 2023). For example, if the goals are spawned only on the black squares of a chessboard during pretraining, then an unseen goal location on one of the white squares can be viewed as an interpolation of the goals in the pretraining tasks. The agent thus may navigate to this unseen test goal by interpolating policies learned from pretraining. In other words, although those evaluation tasks are *unseen* during pretraining, it is not clear whether those tasks can fully demonstrate the challenges of OOD generalization.

To measure the OOD generalization of our proposed safe ICRL methods, we employ a challenging distance-based obstacle and goal spawning strategy: center-oriented for pretraining and edge-oriented for evaluation. The agent spawns at the map center, with obstacles and the goal distributed proportionally closer to this center during pretraining. During evaluation, obstacles and goals follow an edge-oriented distribution. This approach applies similarly to the SafeDarkRoom and SafeDarkMujoco environments. We argue that this setup is more challenging than those in prior studies (Laskin et al., 2023; Zisman et al., 2024; Son et al., 2025), as the agent must learn extrapolation rather than interpolation. More precisely, during pretraining, the grid position $(i, j)$ has an obstacle with probability $\mathbb{P}_{\text{train}}((i, j)) \propto e^{-\alpha d((i,j),c)}$, where $c$ is the map center, $d(\cdot, \cdot)$ denotes Euclidean distance, and $\alpha > 0$ to promote central density.

To generate an evaluation task, the grid position $(i, j)$ has an obstacle with probability $\mathbb{P}_{\text{test}}((i, j)) \propto e^{\alpha d((i,j),c)}$ with $\alpha > 0$ to favor edge density. The goals are generated similarly. This distributional shift is demonstrably OOD, both visually (Appendix C) and mathematically.

**Proposition 1** (Proof in Appendix B.2). *The total variation distance satisfies* $\lim_{\alpha \to \infty} \delta(\mathbb{P}_{train}, \mathbb{P}_{test}) = \lim_{\alpha \to \infty} \frac{1}{2} \sum_O |\mathbb{P}_{train}(O) - \mathbb{P}_{test}(O)| = 1$ *(maximum*

*separation). Similarly, the KL divergence holds* $\lim_{\alpha \to \infty} D_{\text{KL}}(\mathbb{P}_{train} \| \mathbb{P}_{test}) = \infty.$

Proposition 1 shows that as $\alpha \to \infty$, the overlap between the training and test distributions vanishes. As a result, the model must generalize to inputs far outside the regions observed during training.

## 5.4. Analysis

**Question (i).** We now demonstrate emergent safe learning behaviors in the test environments, thus giving an affirmative answer to Question (i). For OOD evaluation, we use edge-oriented obstacles and goals in SafeDarkRoom and SafeDarkMujoco. For unseen ID tasks, we evaluate on SafeVelocity (HalfCheetah and Ant), where target velocities are unseen during training. During evaluation, agents interact with each task for multiple episodes without updating parameters (except for meta-RL baselines, which adapt via parameter updates).

Figure 1 shows SCARED consistently outperforms all baselines by learning safe behavior through in-context interaction, showing increasing episode return and decreasing episode cost as the interaction history grows across all evaluated environments.

Safe AD performs competitively in SafeDarkRoom, showing a similar learning pattern but with slower and weaker adaptation than SCARED. However, its performance degrades as environment complexity increases. In SafetyAnt, episode cost rises while return declines through in-context episodes, and in SafeDarkMujoco, although return improves, cost does not decrease, failing safe adaptation.

Among safe RL baselines, MAML with penalty fails to achieve safe and high-performing adaptation across the evaluated environments. In contrast, SafeMeta achieves higher returns in SafetyVelocity domains but does not reduce safety violations, even when allowed to update parameters during evaluation. This suggests that parameter adaptation alone, without cross-episode interaction histories, is insufficient to regulate safety during online exploration.

**Question (ii).** We demonstrate that the behavior of the pretrained policy can be adjusted at test time by varying CTG, without updating parameters. Intuitively, higher CTG values correspond to greater cost tolerance and enable more aggressive exploration and higher return. Conversely, lower CTG values induce more conservative, safety-focused behavior, potentially at the expense of suboptimal rewards. To study this, we use the same range of CTG introduced before.

SCARED controls CTG exclusively and pursues the maximal achievable return subject to the specified cost budget, avoiding the need to reason about the feasibility of arbitrary RTG-CTG pairs, which is known to be challenging in

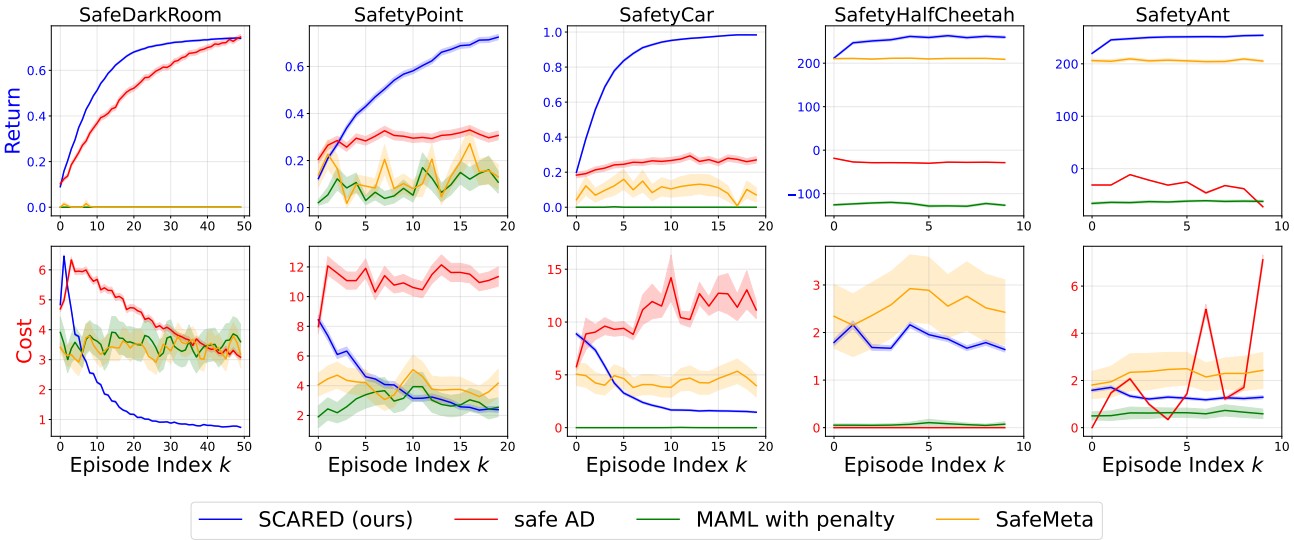

*Figure 1.* **Evaluation performance under increasing in-context episodes.** Curves are averaged over environment-specific OOD evaluation tasks; shaded regions denote standard errors. The $x$-axis shows the episode index $k$, and the $y$-axis reports the episode return $G(\tau_k)$ (top) and episode cost $G_c(\tau_k)$ (bottom). SCARED shows consistent improvement in return and reduction in cost as the in-context interaction grows across all environments, and outperforms all baselines. Safe AD shows environment-dependent performance, failing in the SafeVelocity domain and performing competitively in others, but does not achieve the same performance as SCARED. Safe meta-RL algorithms fail to achieve safe learning behavior due to a lack of in-context interaction, despite updating parameters during evaluation.

constrained settings (Liu et al., 2023). Figure 2 shows that increasing the cost budget consistently leads to higher return, indicating that SCARED learns to map the cost budget to an exploration strategy during in-context adaptation. Notably, maximum episode costs track the CTG target closely, with nearly all episodes respecting the specified budget.

In contrast, Safe AD fails to adapt consistently to varying CTG values, showing inconsistent behavior. For example, in SafeDarkRoom, Safe AD achieves relatively high total return at low CTG targets but shows a downward trend as the cost budget increases, failing to control return-cost tradeoffs. Similarly, safe meta-RL baselines do not show comparable budget-adjustment behavior. Regardless of specified CTG targets, their performance does not vary with the cost budgets, suggesting that parameter adaptation alone does not provide test-time control over reward-cost tradeoffs.

**Ablations.** We conduct ablations on SCARED and Safe AD over shared factors (context length, model size) and method-specific factors (safe AD dataset size), with results reported in Appendix E, revealing distinct sensitivities to these factors. SCARED is largely unaffected by model size, whereas model size significantly influences safe AD performance. For context length, SCARED performs better with longer sequences, while safe AD performs worse. Conversely, safe AD excels with shorter context lengths, while SCARED performs poorly. These results suggest that long-term credit assignment is harder in offline RL due to dataset constraints, while online learning benefits from direct environment interaction. We confirm that Safe AD is highly sensitive to

dataset size: with limited data, it fails to learn and shows random behavior on OOD tasks.

# 6. Related Works

**ICRL.** Learning to improve on a new MDP by interacting with it without parameter updates was first studied in meta-RL (Duan et al., 2016; Wang et al., 2017). See Beck et al. (2025) for a detailed survey of meta-RL. In general, zero-parameter-update generalization in meta-RL is limited, as most methods rely on task identification—matching evaluation tasks to similar pretraining tasks, i.e., identifying pretraining tasks that are similar to the evaluation task and acting as if the evaluation task were the pretraining tasks. Laskin et al. (2023) coined the term ICRL and demonstrated strong generalization to tasks far from the pretraining distribution, spurring growing interest in the area.

Variations of algorithm distillation have been proposed for efficient in-context reinforcement learning, including the Decision Pretrained Transformer (Lee et al., 2023) and Algorithm Distillation with Noise (Zisman et al., 2024). Both approaches aim to efficiently train transformer models for in-context learning using optimal policies. LLMs have been shown to exhibit ICRL as an emergent capability, enabling them to keep improving on tasks when previous attempts are given along with a reward in the context (Song et al., 2026). See Moeini et al. (2025) for a detailed survey of ICRL. However, none of these works have investigated safety within the context of ICRL.

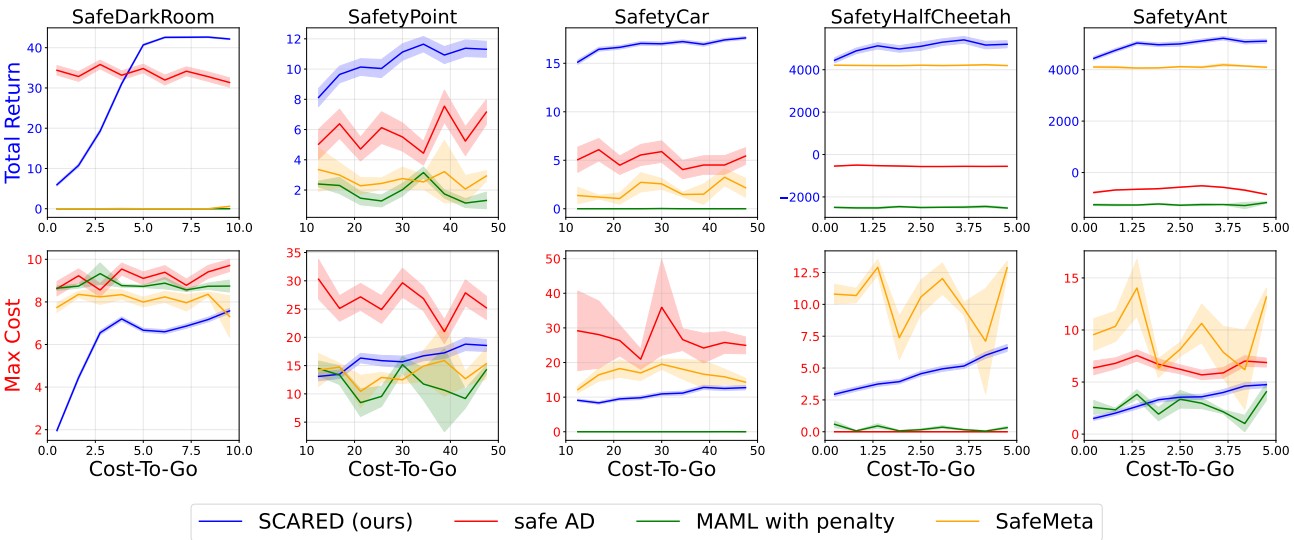

*Figure 2.* **Evaluation performance under varying CTG targets.** Results are averaged over environment-specific OOD evaluation tasks; shaded regions denote standard errors. The $x$-axis shows the target CTG, and the $y$-axis reports the total return $\sum_{k=1}^{K} G(\tau_k)$ (top) and the maximum episode cost $\max_{k \in [1,K]} G_c(\tau_k)$ (bottom). SCARED shows a clear pattern: as the cost budget increases, it achieves higher total return across all environments. Other algorithms fail to present this budget-adjustment behavior.

**Safe RL.** Safe RL methods commonly adopt CMDP formulations to enforce compliance with safety constraints during exploration and policy optimization (Garcıa & Fernández, 2015; Gu et al., 2024; Wachi et al., 2024). Constrained Policy Optimization (CPO) serves as a foundational algorithm in this area, balancing rewards with safety (Achiam et al., 2017; Wachi & Sui, 2020). Building on CPO, subsequent works enhance theoretical guarantees and scalability through primal-dual formulations and efficient projection-based updates (Tessler et al., 2019; Xu et al., 2021; Yang et al., 2022). Shielding with function encoders and conformal prediction has been proposed to handle unseen OOD environments (Kwon et al., 2025). However, this work emphasizes runtime adaptation rather than learning algorithms through interaction. From the offline RL perspective, Constrained Decision Transformer (Liu et al., 2023) uses Transformer architecture for safe RL. However, existing approaches enforce safety constraints only when the evaluation task closely matches the pretraining tasks. In contrast, our safe ICRL framework satisfies safety constraints even on evaluation tasks that differ substantially from the pretraining distribution.

**Safe Meta RL.** Safe meta RL enables fast adaptation to new tasks, enforcing safety constraints. For example, Luo et al. (2021) propose a three-phase strategy that meta-learns a safety critic across environments, adapts it to new tasks using limited offline data, and use it with a recovery policy to reduce constraint violations during learning. Khattar et al. (2023) provides task-averaged regret bounds for rewards and constraints via gradient-based meta-learning. More recently, Guan et al. (2024) proposes a cost-aware context encoder

that uses supervised cost relabeling and contrastive learning to infer tasks based on safety constraints. Xu & Zhu (2025) proposes a safe meta-RL algorithm that integrates a closed-form, one-step safe policy adaptation mechanism to provide anytime safety guarantees. Unlike safe meta-RL methods, which rely on parameter updates to achieve safe adaptation during evaluation, safe ICRL enables adaptation without test-time parameter updates.

# 7. Conclusion

This work pioneers the study of safe ICRL, where agents must adapt to new and potentially OOD tasks without parameter updates while adhering to safety constraints. We show that existing approaches, including algorithm distillation for safe RL, and safe meta-RL baselines, are limited in this setting, as they lack mechanisms for regulating safety through in-context interaction histories and for adjusting to cost budgets during adaptation.

To address these limitations, we introduced SCARED, a re-inforcement pretraining approach for safe in-context adaptation under the CMDP framework. SCARED enables agents to balance reward maximization and cost minimization during adaptation by regulating episode-level safety constraints and adjusting behavior in response to different cost budgets. Theoretically, we prove that fixed points of SCARED optimization satisfy strict safety constraints while maximizing rewards. To rigorously test OOD generalization, we introduce challenging benchmarks for safe ICRL (Proposition 1), addressing a gap in prior evaluations (Laskin et al., 2023; Zisman et al., 2024; Son et al., 2025). Through these chal-

lenging benchmarks, we confirm that our pretrained safe ICRL agents adapt to not only unseen ID but also OOD evaluation tasks while respecting safety constraints. We envision this work guiding the development of robust, generalizable, and safety-aware in-context learning methods for real-world applications.

## Impact Statement

This work introduces the first study of safety in ICRL, a paradigm in which agents adapt to new tasks at deployment time without parameter updates. By providing a framework for enforcing user-specified safety budgets during test-time adaptation, this work is relevant to real-world applications such as robotics and embodied AI, where retraining is impractical and unsafe exploration can have real consequences.

Our method applies to problems that can first be modeled as CMDPs, which requires available cost signals and an appropriate CTG budget; selecting this budget may require domain knowledge or calibration. The proposed framework does not eliminate risk, particularly in environments with unmodeled hazards or adversarial conditions, and should be complemented by human oversight and additional safety mechanisms. We release our code, benchmarks, and evaluation protocols[2] to support transparent and reproducible research on safe adaptation under distribution shift.

## Acknowledgments

This work is supported in part by the US National Science Foundation under the awards CCF-1942836, III-2128019, SLES-2331904, and CAREER-2442098, the Commonwealth Cyber Initiative's Central Virginia Node under the award VV-1Q26-001, a Cisco Faculty Research Award, and an Nvidia academic grant program award.

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

# A. Algorithm

---

**Algorithm 1** SCARED Implemented with DDPG

---

1: **Input:** discount factor $\gamma$, cost budget $\delta$, number of training steps $T_{\max}$, batch size $N$, env time limit $t_{\max}$, episodes-per-history range $[K_{\min}, K_{\max}]$, CTG range $[\text{CTG}_{\min}, \text{CTG}_{\max}]$, environment distribution $\mathcal{E}$

2: **Initialize:** actor $\pi(\cdot \mid s, H, \text{CTG}; \theta_p)$, reward critic $Q(s, a; \theta_v)$, cost critic $Q^c(s, a; \theta_c)$, and target nets $\{\pi', Q', Q^{c'}\}$. $\lambda \leftarrow 0$,    replay buffer $R \leftarrow \varnothing$

3: **for** $T \leftarrow 1$ **to** $T_{\max}$ **do**

4:     $K \leftarrow \text{rnd}(K_{\min}, K_{\max})$,    $\text{CTG} \leftarrow \text{rnd}(\text{CTG}_{\min}, \text{CTG}_{\max})$,    $t \leftarrow 0$
       $H \leftarrow [\,]$    *// list of episodes*

5:     sample $\text{env} \sim \mathcal{E}$,   $s_t \leftarrow \texttt{env.reset()}$

6:     **for** $k \leftarrow 1$ **to** $K$ **do**

7:         $t_{\text{start}} \leftarrow t$,    $\text{CTG}_k \leftarrow \text{CTG}$,    $e_k \leftarrow [\,]$    *// list of transitions*

8:         **while** $t - t_{\text{start}} < t_{\max}$ **and** $s_t$ not terminal **do**

9:             sample $a_t \sim \pi_{\theta_p}(\cdot \mid s_t, H, \text{CTG}_k)$

10:             step $a_t$ in env $\to (s_{t+1}, r_{t+1}, c_{t+1})$

11:             append $(s_t, a_t, r_{t+1}, c_{t+1}, s_{t+1}, \text{CTG}_k)$ to $e_k$

12:             $\text{CTG}_k \leftarrow \text{CTG}_k - c_{t+1}$,    $t \leftarrow t + 1$

13:         **end while**

14:         append $e_k$ to $H$

15:         $s_t \leftarrow \texttt{env.reset()}$

16:     **end for**

17:     append $H$ to $R$

18:     $\mathcal{D} \leftarrow \texttt{Sample}(R, N)$    *// N trajectories*

19:     reset accumulators: $d\theta_v \leftarrow 0, d\theta_c \leftarrow 0, d\theta_p \leftarrow 0, d\lambda \leftarrow 0$

20:     **for each** trajectory $H \in \mathcal{D}$ **do**

21:         **for each** episode $e \in H$ **do**

22:             $C_e \leftarrow \sum_{c \in e} c$,    $C_e^{\max} \leftarrow -\infty$,    $v_e \leftarrow \mathbf{1}\{C_e > \delta\}$

23:             **for each** $(s_t, a_t, r_{t+1}, c_{t+1}, s_{t+1}, \text{CTG}_t) \in e$ **do**

24:                 sample $a'_{t+1} \sim \pi_{\theta'_p}(\cdot \mid s_{t+1}, H_{\leq t+1}, \text{CTG}_t - c_{t+1})$

25:                 $L_v \leftarrow \big(Q_{\theta_v}(s_t, a_t) - [\,r_{t+1} + \gamma Q_{\theta'_v}(s_{t+1}, a'_{t+1})\,]\big)^2$

26:                 $d\theta_v \leftarrow d\theta_v + \nabla_{\theta_v} L_v$

27:                 $L_c \leftarrow \big(Q^c_{\theta_c}(s_t, a_t) - [\,c_{t+1} + \gamma Q^c_{\theta'_c}(s_{t+1}, a'_{t+1})\,]\big)^2$

28:                 $d\theta_c \leftarrow d\theta_c + \nabla_{\theta_c} L_c$

29:                 sample $\hat{a}_t \sim \pi_{\theta_p}(\cdot \mid s_t, H_{\leq t}, \text{CTG}_t)$

30:                 $L_p \leftarrow -Q_{\theta_v}(s_t, \hat{a}_t) + \lambda v_e Q^c_{\theta_c}(s_t, \hat{a}_t)$

31:                 $d\theta_p \leftarrow d\theta_p + \nabla_{\theta_p} L_p$

32:             **end for**

33:             $C_e^{\max} \leftarrow \max\{C_e^{\max}, C_e\}$

34:         **end for**

35:         $d\lambda \leftarrow d\lambda + (C_e^{\max} - \delta)$

36:     **end for**

37:     apply parameter updates to $\theta_v, \theta_c, \theta_p, \lambda$ using $d\theta_v, d\theta_c, d\theta_p, d\lambda$

38:     update targets $\pi', Q', Q^{c'}$

39: **end for**

---

Note that our proposed optimization (6) can be used with any policy optimizer. In our implementation, we chose DDPG following Grigsby et al. (2024a) to support continuous and discrete action spaces and to enable highly parallel data-collection while doing off-policy training.

# B. Proofs

We organize the proofs into two subsections: one for Theorem 1 and one for Proposition 1. For Theorem 1, we first prove the supporting lemma and then Theorem 1 itself. For Proposition 1, we provide proofs for both Total Variation Distance and KL Divergence.

## B.1. Proof of Theorem 1

**Lemma 1.** *Let* $[x]_+ = \max\{0, x\}$. *Fix* $(\bar{\lambda}, \bar{s}) \in \mathbb{R} \times \mathbb{R}$. *If for all sufficiently small* $\eta > 0$, $\bar{\lambda} = [\bar{\lambda} + \eta\,\bar{s}]_+$, *then* $\bar{\lambda} \geq 0$, $\bar{s} \leq 0$, *and* $\bar{\lambda}\,\bar{s} = 0$.

*Proof.* Consider two cases.

*Case* $\bar{\lambda} > 0$. For all small $\eta > 0$, $\bar{\lambda} + \eta\bar{s} > 0$, hence

$$[\bar{\lambda} + \eta\bar{s}]_+ = \bar{\lambda} + \eta\bar{s}.$$

The equality $\bar{\lambda} = [\bar{\lambda} + \eta\bar{s}]_+$ then forces $\bar{\lambda} = \bar{\lambda} + \eta\bar{s}$, so $\bar{s} = 0$. Thus $\bar{s} \leq 0$ and $\bar{\lambda}\bar{s} = 0$ hold.

*Case* $\bar{\lambda} = 0$. Then we have $0 = [\eta\bar{s}]_+ = \max\{0, \eta\bar{s}\}$ for all small $\eta > 0$, which implies $\eta\bar{s} \leq 0$, hence $\bar{s} \leq 0$. Trivially $\bar{\lambda}\bar{s} = 0$ and $\bar{\lambda} \geq 0$. $\square$

We now proceed to the proof of Theorem 1.

*Proof.* Let $J(\pi) := \mathbb{E}_\pi\left[\sum_{k=1}^K G(\tau_k)\right]$ in this proof. We prove the theorem in both directions. Also let $g_k^+(\pi) := \max\{0, g_k(\pi)\}$, and $s(\pi) := \max_k g_k(\pi)$.

*1. Fixed point $\Rightarrow$ Primal optimal.*

*Feasibility.* By the lemma 1, $s(\bar{\pi}) \leq 0$. Since $s(\bar{\pi}) = \max_i g_i(\bar{\pi})$, each $g_i(\bar{\pi}) \leq 0$. Thus $\bar{\pi}$ is feasible.

*Optimality.* Let $\pi^\star$ be any optimal feasible policy (exists by assumption 1). On the feasible set, $g_i^+(\pi^*) = 0$, so $L_\Sigma(\pi^\star, \bar{\lambda}) = J(\pi^\star)$. Because $\bar{\pi}$ maximizes $L_\Sigma(\cdot, \bar{\lambda})$,

$$J(\bar{\pi}) = L_\Sigma(\bar{\pi}, \bar{\lambda}) \geq L_\Sigma(\pi^\star, \bar{\lambda}) = J(\pi^\star).$$

Conversely, by optimality of $\pi^\star$ among feasible policies and feasibility of $\bar{\pi}$, $J(\bar{\pi}) \leq J(\pi^\star)$. Hence $J(\bar{\pi}) = J(\pi^\star)$.

*2. Primal optimal $\Rightarrow$ Fixed point.*

Let $\lambda^\star$ be an optimal dual solution (assumption 2). Then by strong duality,

$$p^\star = \max_\pi \left(J(\pi) - \sum_i \lambda_i^\star g_i(\pi)\right).$$

Therefore, for every $\pi$,

$$J(\pi) - \sum_i \lambda_i^\star g_i(\pi) \leq p^\star. \tag{7}$$

Using $\lambda_i^\star \leq \|\lambda^\star\|_\infty$ and $g_i^+(\pi) \geq g_i(\pi)$,

$$\sum_i \lambda_i^\star g_i(\pi) \leq \|\lambda^\star\|_\infty \sum_i g_i^+(\pi).$$

Subtract this from $J(\pi)$ and combine with (7):

$$
\begin{aligned}
L_\Sigma\big(\pi, \|\lambda^\star\|_\infty\big) &= J(\pi) - \|\lambda^\star\|_\infty \sum_i g_i^+(\pi) \\
&\leq J(\pi) - \sum_i \lambda_i^\star g_i(\pi) \\
&\leq p^\star. \tag{8}
\end{aligned}
$$

For a primal-optimal $\pi^\star$, feasibility gives $\sum_i g_i^+(\pi^\star) = 0$, hence

$$L_\Sigma\big(\pi^\star, \|\lambda^\star\|_\infty\big) = J(\pi^\star) = p^\star.$$

Together with (8), this shows

$$\pi^\star \in \arg\max_\pi L_\Sigma\big(\pi, \|\lambda^\star\|_\infty\big).$$

It remains to check multiplier stationarity at $\bar{\lambda} = \|\lambda^\star\|_\infty$. By complementary slackness, $\sum_i \lambda_i^\star g_i(\pi^\star) = 0$. If $\|\lambda^\star\|_\infty > 0$, at least one constraint is active at $\pi^\star$, so $\max_i g_i(\pi^\star) = 0$. Hence $[\bar{\lambda} + \eta\,s(\pi^\star)]_+ = [\bar{\lambda} + \eta \cdot 0]_+ = \bar{\lambda}$ for all small $\eta > 0$. If $\|\lambda^\star\|_\infty = 0$, $\pi^\star$ is unconstrained-optimal with $s(\pi^\star) \leq 0$, and $[0 + \eta\,s(\pi^\star)]_+ = 0$. In both cases the projection is stationary. Thus $(\pi^\star, \bar{\lambda})$ with $\bar{\lambda} = \|\lambda^\star\|_\infty$ is a fixed point.

*On the multiplier update.*
If $s(\pi_{t+1}) > 0$, then $\lambda_{t+1} = [\lambda_t + \eta s(\pi_{t+1})]_+ \geq \lambda_t$. If constraints are strict ($s(\pi_{t+1}) < 0$), then $\lambda_{t+1} \leq \lambda_t$. Under persistent violation, $\lambda_t$ eventually exceeds $\|\lambda^*\|_\infty$, by the exact-penalty bound (8), any maximizer of $L_\Sigma(\cdot, \lambda_t)$ is then feasible/optimal and $s(\pi_{t+1}) \leq 0$, so $\lambda$ stabilizes. $\square$

Note: The full convergence proof to an exact fixed point is beyond the scope of this work.

## B.2. Proof of Proposition 1

**Total Variation Distance** First, we present the exact form of probabilities for $\mathbb{P}_{\text{train}}(O)$ and $\mathbb{P}_{\text{test}}(O)$ where $O \in \mathcal{O} = \{(i,j)\}_{1 \leq i \leq n, 1 \leq j \leq m}$. Let the map of center $c = (i_c, j_c)$. Then, $\mathbb{P}_{\text{train}}((i,j)) \propto e^{-\alpha d((i,j),c)}$ and $\mathbb{P}_{\text{train}}((i,j)) \propto e^{-\alpha d((i,j),c)}$ implies:

$$\mathbb{P}_{\text{train}}((i,j)) = \frac{e^{-\alpha((i-i_c)^2+(j-j_c)^2)}}{Z_{-\alpha}} \quad \text{and}$$

$$\mathbb{P}_{\text{test}}((i,j)) = \frac{e^{\alpha((i-i_c)^2+(j-j_c)^2)}}{Z_{\alpha}}$$

, where $Z_{-\alpha} = \sum_{(i',j') \in \mathcal{O}} e^{-\alpha((i'-i_c)^2+(j'-j_c)^2)}$ and $Z_{\alpha} = \sum_{(i',j') \in \mathcal{O}} e^{\alpha((i'-i_c)^2+(j'-j_c)^2)}$. Let us define $supp(P)$ as the set $\{x \in Dom(P) \mid P(x) > 0\}$. If $supp(\mathbb{P}_{\text{train}}) \cap supp(\mathbb{P}_{\text{test}}) = \emptyset$ holds, then for each $O \in \mathcal{O}$, $\mathbb{P}_{\text{test}}(O) > 0$ implies $\mathbb{P}_{\text{train}}(O) = 0$, and $\mathbb{P}_{\text{train}}(O) > 0$ implies $\mathbb{P}_{\text{test}}(O) = 0$. Hence, we obtain

$$\delta(\mathbb{P}_{\text{train}}, \mathbb{P}_{\text{test}}) = \frac{1}{2} \sum_{O \in \mathcal{O}} |\mathbb{P}_{\text{train}}(O) - \mathbb{P}_{\text{test}}(O)| \quad (9)$$

$$= \frac{1}{2} \sum_{O \in \mathcal{O}} (|\mathbb{P}_{\text{train}}(O)| + |\mathbb{P}_{\text{test}}(O)|) = 1. \quad (10)$$

Now, we claim that $supp(\mathbb{P}_{\text{train}}) \cap supp(\mathbb{P}_{\text{test}}) = \emptyset$ as $\alpha \to \infty$. For the training distribution $\mathbb{P}_{\text{train}}$, the term $e^{-\alpha\left((i-i_c)^2+(j-j_c)^2\right)} \to 0$ if $(i-i_c)^2 + (j-j_c)^2 > 0$. Hence,

$$Z_{-\alpha} = e^{-\alpha \cdot 0} + \sum_{(i,j) \neq (i_c,j_c)} e^{-\alpha\left((i-i_c)^2+(j-j_c)^2\right)} \to 1$$

Hence, only when $i = i_c$ and $j = j_c$, we have positive probability $\mathbb{P}_{\text{train}}((i_c, j_c)) = \frac{e^{-\alpha \cdot 0}}{Z_{-\alpha}} = \frac{1}{Z_{-\alpha}} = 1$. Thus $supp(\mathbb{P}_{\text{train}}) \to \{(i_c, j_c)\}$

For the test distribution $\mathbb{P}_{\text{test}}$, the term $e^{\alpha\left((i-i_c)^2+(j-j_c)^2\right)}$ is maximized when $(i-i_c)^2 + (j-j_c)^2$ is maximized. Let $d_{\max} = \max_{(i,j) \in \mathcal{O}} \left((i-i_c)^2 + (j-j_c)^2\right)$, and $\mathcal{O}_{\text{edge}} = \left\{(i,j) \in \mathcal{O} \mid (i-i_c)^2 + (j-j_c)^2 = d_{\max}\right\}$. Then the partial function can be decomposed into two terms:

$$Z_{\alpha} = \sum_{(i,j) \in \mathcal{O}_{\text{edge}}} e^{\alpha d_{\max}} + \sum_{(i,j) \notin \mathcal{O}_{\text{edge}}} e^{\alpha\left((i-i_c)^2+(j-j_c)^2\right)}.$$

Hence, as $\alpha \to \infty$, $(i-i_c)^2 + (j-j_c)^2 < d_{\max}$ implies that $\frac{e^{\alpha((i-i_c)^2+(j-j_c))}}{Z_{\alpha}} \to 0$. Thus, we obtain

$$\mathbb{P}_{\text{test}}((i,j)) \to \begin{cases} \frac{1}{|\mathcal{O}_{\text{edge}}|} & \text{if } (i,j) \in \mathcal{O}_{\text{edge}}, \\ 0 & \text{otherwise} \end{cases} \quad as \quad \alpha \to \infty.$$

Hence, $supp(\mathbb{P}_{\text{test}}) \to \mathcal{O}_{\text{edge}}$. Finally, we conclude that $supp(\mathbb{P}_{\text{train}}) \cap supp(\mathbb{P}_{\text{test}}) = \{(i_c, j_c)\} \cap \mathcal{O}_{\text{edge}} = \emptyset$ as $\alpha \to \infty$.

**KL Divergence** We keep the same notation to the proof so far. By the definition of KL divergence and explicit form of the probabilities, we have

$$D_{\text{KL}}(\mathbb{P}_{\text{train}} \| \mathbb{P}_{\text{test}})$$

$$= \sum_{(i,j) \in \mathcal{O}} \mathbb{P}_{\text{train}}((i,j)) \log \frac{\mathbb{P}_{\text{train}}((i,j))}{\mathbb{P}_{\text{test}}((i,j))}$$

$$= \sum_{(i,j) \in \mathcal{O}} \mathbb{P}_{\text{train}}((i,j)) \log \frac{Z_{\alpha} e^{-\alpha((i-i_c)^2+(j-j_c)^2)}}{Z_{-\alpha} e^{\alpha((i-i_c)^2+(j-j_c)^2)}}$$

$$= \sum_{(i,j) \in \mathcal{O}} \mathbb{P}_{\text{train}}((i,j)) \left(\log \frac{Z_{\alpha}}{Z_{-\alpha}} - 2\alpha((i-i_c)^2 + (j-j_c)^2)\right). \quad (11)$$

As we have shown in the previous proof for TV distance, $Z_{\alpha} \to |\mathcal{O}_{\text{edge}}| e^{\alpha d_{\max}}$ and $Z_{-\alpha} \to 1$. Hence, for pairs $(i,j) \neq (i_c, j_c)$, the associated term $\mathbb{P}_{\text{train}}((i,j)) \left(\log \frac{Z_{\alpha}}{Z_{-\alpha}} - 2\alpha((i-i_c)^2 + (j-j_c)^2)\right)$ goes to $\mathbb{P}_{\text{train}}((i,j)) (\log |\mathcal{O}_{\text{edge}}| + \alpha d_{\max} - 2\alpha d((i,j),c))$. For $(i,j) \neq (i_c, j_c)$ and $d_{\min} = \min_{(i,j) \neq (i_c,j_c)} d((i,j),c)$, we have

$$P_{\text{train}}((i,j)) = \frac{e^{-\alpha d((i,j),c)}}{Z_{-\alpha}} \leq e^{-\alpha d_{\min}} \to 0. \quad (12)$$

Since the exponential rate converges to zero more rapidly than the linear rate with respect to $\alpha$, the term $\mathbb{P}_{\text{train}}((i,j)) (\log |\mathcal{O}_{\text{edge}}| + \alpha d_{\max} - 2\alpha d((i,j),c))$ goes to 0 as $\alpha \to \infty$. Therefore, we only consider the term when $(i,j) = (i_c, j_c)$. But this term comes down to $\mathbb{P}_{\text{train}}((i,j)) \log \frac{Z_{\alpha}}{Z_{-\alpha}}$, which goes to $\infty$ as $\alpha \to \infty$. Thus $D_{\text{KL}}(\mathbb{P}_{\text{train}} \| \mathbb{P}_{\text{test}}) \geq \infty$.

## C. Training Details

**Environments.** We introduce the details of the environments. The environments are visualized in Figure 3. Both environments use $\alpha = 0.5$ to generate goals and obstacles.

For SafeDarkRoom, we use a 9x9 Grid and give one reward upon reaching the goal and incur one cost when going on an obstacle cell. We terminate the episode whenever the agent reaches the goal, which results in a truly sparse reward, often referred to as *DarkRoom Hard*.

For SafeDarkMujoco, a continuous state and action space counterpart to SafeDarkRoom, the agent lacks lidar information and is blind to goal and obstacle locations. Instead, it perceives its own position and rotation matrix. A sparse reward is obtained when the agent reaches the goal, terminating the episode. To enhance runtime efficiency, we

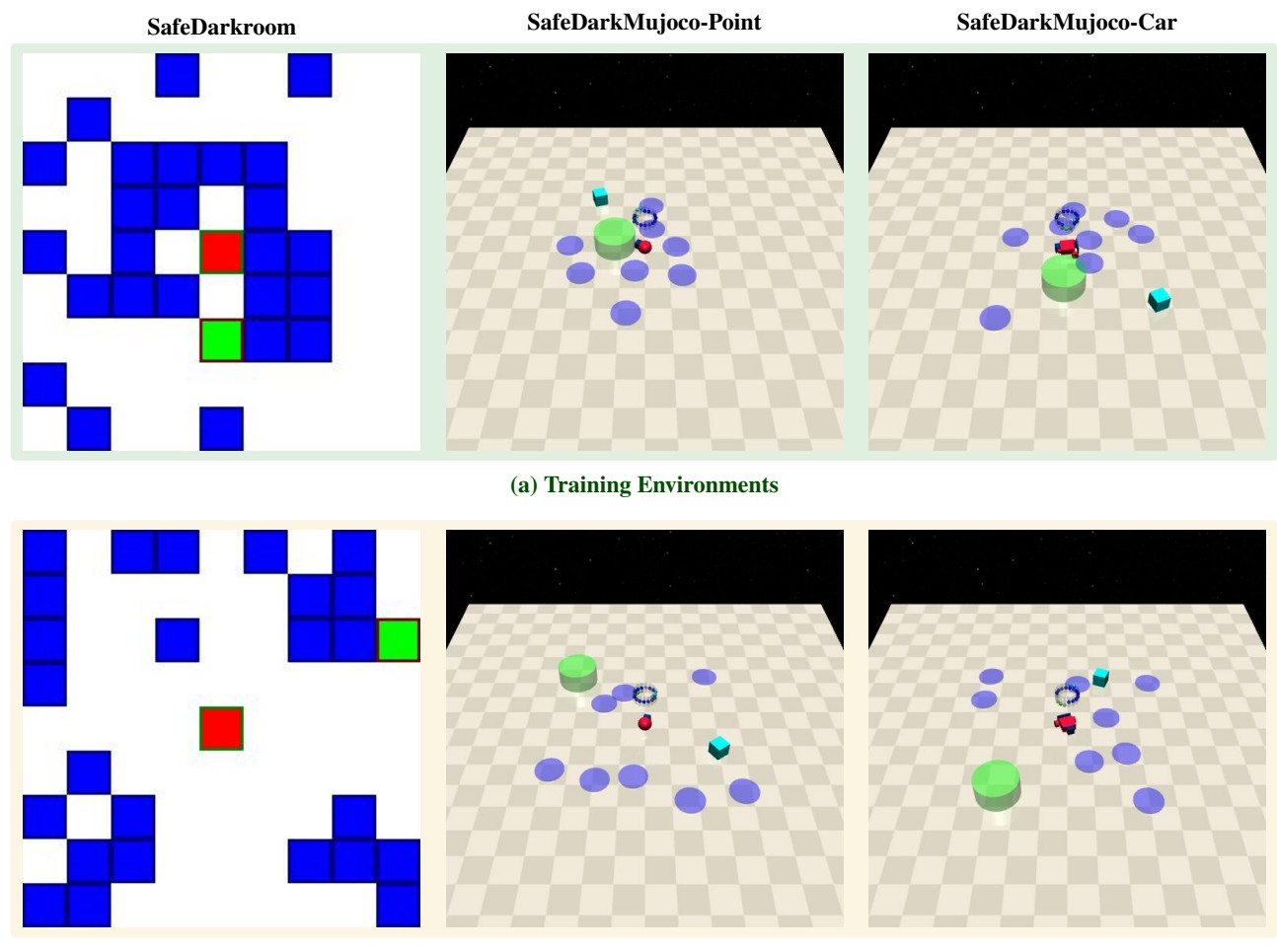

(a) Training Environments

(b) Test Environments

*Figure 3.* During training, goals and obstacles are generated with a center-oriented approach, while during evaluation, they are edge-oriented. This applies consistently to both goals and obstacles. We set $\alpha = 0.5$ for generating goals and obstacles. The red color denotes the robot, the green color represents the goal location, and obstacles are depicted in shades of blue.

employ macro actions, compressing $n$ simulation steps into a single step. For example, with $n = 5$, a policy action is repeated over five internal simulation steps, reducing the default 250 simulation steps to $\frac{250}{n}$. In our experiments, we set $n = 5$.

For SafeVelocity, we use the Ant and HalfCheetah environments from SafetyGym (Ji et al., 2023), but sample a different target velocity for each trajectory to convert them into safe ICRL benchmarks. Other aspects of the environments remain unchanged.

**SCARED.** While running Algorithm 1 for reinforcement pretraining, we resample a new environment from the training distribution described in Section 5 every $K$ episodes. For each environment, a CTG is also sampled, ranging from $[1, 15]$ for SafeDarkRoom and $[10, 50]$ for SafeDarkMujoco. The remaining hyperparameters are provided in Table 2.

Our architecture follows Grigsby et al. (2024a). We em-

ploy an MLP time-step encoder that maps each tuple $(S_t, A_t, R_t, C_t)$ to an embedding, which is then fed into a transformer-based trajectory encoder. A prediction head outputs either the action distribution (for discrete actions) or the value (for continuous actions).

**SafeMeta & MAML with penalty** We use the original implementations by (Xu & Zhu, 2025) with 3 hidden layers of (64, 512, 64) units for policy, value, and cost networks, totaling 205,510 parameters, comparable to SCARED. We train for 15k meta-iterations, where each iteration samples 20 tasks from the training distribution. During meta-testing, both algorithms perform $K$ gradient updates to adapt to new tasks with unseen cost-to-go values. Note that while some meta-RL methods use frame stacking or recurrent architectures, their hidden states reset between episodes, making them fundamentally episodic—they do not carry information across episode boundaries as ICRL methods do.

| Domain | Training tasks | Evaluation tasks | Evaluation CTG targets |
|---|---|---|---|
| SafeDarkRoom | Center-oriented goals and obstacles with $\alpha = 0.5$ | Edge-oriented goals and obstacles with $\alpha = 0.5$ | Uniform over $[1, 10]$; 100 targets |
| SafeDarkMujoco (Point, Car) | Center-oriented goals and obstacles with $\alpha = 0.5$ | Edge-oriented goals and obstacles with $\alpha = 0.5$ | Uniform over $[10, 50]$; 100 targets |
| SafeVelocity (HalfCheetah, Ant) | Target-velocity tasks excluding the held-out multipliers | Held-out target-velocity multipliers sampled from $[0.5, 1.0]$ in increments of 0.05 | Uniform over $[0, 5]$; 100 targets |

*Table 1.* Summary of the training and evaluation setup used across domains. The CTG ranges and sample count follow the evaluation setup in Section 5; the same CTG intervals and sample counts are used as cost budgets for SafeMeta and MAML with penalty, while Safe AD uses paired RTG-CTG targets with $\mathrm{CTG_{max}} = 10, 50, 5$ for SafeDarkRoom, SafeDarkMujoco, and SafeVelocity, respectively.

| Parameter | SafeDarkRoom | SafeDarkMujoco & SafeVelocity |
|---|---|---|
| $K_{\min}, K_{\max}$ | 50, 50 | 20, 20 |
| Episode time limit $t_{\max}$ | 30 | 75 |
| Replay buffer capacity | 100,000 | 100,000 |
| Embedding Dim | 64 | 64 |
| Hidden Dim | 64 | 64 |
| Num Layers | 4 | 4 |
| Num Heads | 8 | 8 |
| Seq Len | 1500 | 1500 |
| Attention Dropout | 0 | 0 |
| Residual Dropout | 0 | 0 |
| Embedding Dropout | 5 | 5 |
| Learning Rate | 3e-4 | 3e-4 |
| Betas | (0.9, 0.99) | (0.9, 0.99) |
| Clip Grad | 1.0 | 1.0 |
| Batch Size | 32 | 32 |
| Num Updates | 30k | 10k |
| Optimizer | Adam | Adam |

*Table 2.* Parameters for SCARED

**Safe Algorithm Distillation.** In safe algorithm distillation, we collect a dataset $\mathcal{D} = \{\Xi_i\}$ comprising multiple trajectories, where each trajectory $\Xi_i \doteq (\tau_1, \tau_2, \ldots, \tau_K)$ represents a sequence of episodes generated by running existing safe RL algorithms on various CMDPs. Each episode $\tau_k$ in a trajectory $\Xi_i$ comprises states, actions, rewards, and costs, with the episode return $G(\tau_k) \doteq \sum_{t=1}^{T} R_t$ and cost-to-go $G_{c,t}(\tau_k) \doteq \sum_{i=t+1}^{T} C_i$, expected to increase and decrease, respectively, with $k$ as the RL algorithm learns.

We train the policy $\pi_\theta$ autoregressively using safe algorithm distillation to distill the behavior of safe RL algorithms present in the dataset, following Laskin et al. (2023). Concretely, each training example consists of a trajectory segment $\Xi_i$ containing state and action pairs along with RTG and CTG conditioning. Let the policy take as input $(S_t^k, H_t^k, G_t(\tau_k), G_{c,t}(\tau_k))$ and predict the action at time $t$ for episode $k$.

**Discrete action spaces.** For environments with discrete actions (e.g., SafeDarkRoom), the model outputs categorical logits $z_\theta(\cdot)$, generating a categorical distribution $\pi_\theta(\cdot \mid \cdot) = \mathrm{Softmax}(z_\theta(\cdot))$. We train with the standard cross-entropy loss:

$$\mathcal{L}_{\mathrm{disc}}(\theta) = \mathbb{E}_{\Xi_i \sim \mathcal{D}} \left[ -\log \pi_\theta\left(A_t^k \mid S_t^k, H_t^k, G_t(\tau_k), G_{c,t}(\tau_k)\right) \right]. \tag{13}$$

**Continuous action spaces.** For environments with continuous actions (e.g., SafeDarkMujoco), the model outputs a conditional distribution over actions parameterized by a mean $\mu_\theta(\cdot)$. We supervise the predicted mean using an $\ell_2$ loss:

$$\mathcal{L}_{\mathrm{cont}}(\theta) = \mathbb{E}_{\Xi_i \sim \mathcal{D}} \left[ \left\| A_t^k - \mu_\theta\left(S_t^k, H_t^k, G_t(\tau_k), G_{c,t}(\tau_k)\right) \right\|_2^2 \right]. \tag{14}$$

These objectives enables the transformer to distill constraint-aware goal seeking behaviors into its forward pass.

**Dataset Collection.**

| Parameter | SafeDarkRoom | SafeDarkMujoco & SafeVelocity |
|---|---|---|
| $K_{\min}, K_{\max}$ | 50, 50 | 20, 20 |
| Episode time limit $t_{\max}$ | 30 | 75 |
| Hidden layers | (64, 512, 64) | (64, 512, 64) |
| Min/Max batch size | 500/1500 | 1500/1500 |
| Policy learning rate | 1e-3 | 1e-3 |
| Value/Cost learning rate | 3e-2/1e-1 | 3e-2/1e-1 |
| Discount factor $\gamma$ | 0.99 | 0.99 |
| GAE parameter $\tau$ | 0.95 | 0.95 |
| Max KL divergence | 1e-3 | 1e-3 |
| Lagrangian weight $\lambda$ | 1.0 | 1.0 |
| Num meta-training iterations | 15k | 15k |
| Policy optimizer | Adam | Adam |
| Value/Cost optimizer | L-BFGS | L-BFGS |

*Table 3.* Parameters for SafeMeta and MAML with penalty

For safe AD, we collect learning trajectories using safe RL algorithms. As our base safe RL algorithm, we employ PPO-Lagrangian (Schulman et al., 2017; Ray et al., 2019), which is designed to maximize rewards while enforcing safety constraints. These trajectories capture behaviors that learn to avoid obstacles.

To introduce variation in the learned behaviors, we vary the cost limits in PPO-Lag across multiple settings. For the SafeDarkRoom environment, we use three cost limits: 0, 2.5, and 5.0. For each cost limit, we collect 50,000 steps of learning history. For the SafeDarkMujoco and SafetyVelocity environments, prior offline RL benchmarks and constrained decision transformer methods typically use datasets on the order of 1–2 million environment steps (Fu et al., 2020; Liu et al., 2023). In our experiments, we intentionally use larger datasets of 2–5 million steps. This difference from the usual convention is motivated by the need to collect diverse cost-related experiences in continuous control settings, where meaningful safety-constrained behavior emerges only after long training horizons.

In our data collection, we set a single cost limit of 0 throughout the full learning history. This setup intentionally includes trajectories generating a various range of cost violations, from high-cost exploration phases to near-zero-cost behavior as training progresses. Continuous control tasks are substantially more difficult to solve than discrete ones, and learning meaningful policies under safety constraints typically requires more interaction data.

Thus, this design poses a challenge for safe supervised in-context RL: conditioning on both return-to-go and cost-to-go induces a large combinatorial space of reward–cost trade-offs that must be represented in the dataset. As a result, the required dataset size grows substantially.

**Full-pipeline budget comparison.** Safe AD requires both an offline data-collection stage and a subsequent distillation stage, whereas SCARED trains directly through online reinforcement pretraining. Under full-pipeline accounting, Safe AD uses more interaction and compute than SCARED. For example, on SafeDarkRoom, Safe AD uses about 4M environment steps including PPO-Lagrangian data collection, compared with about 1M environment steps for SCARED. Safe AD also uses a larger transformer, with about 25M parameters compared with about 6M parameters for SCARED.

**Hyper Parameters.** We report the hyperparameters used for SCARED (Table 2) and safe AD (Table 4). SCARED adopts the AMAGO framework (Grigsby et al., 2024a) integrated with our SCARED method. Safe AD employs a constrained decision transformer as the backbone (Liu et al., 2023).

## D. Variation of Safe Algorithm Distillation

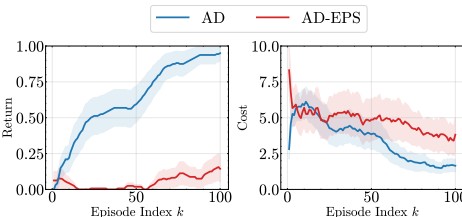

*Figure 4.* Performance comparison between algorithm distillation and algorithm distillation with noise in SafeDarkRoom. AD-EPS fails to generalize to out-of-distribution (OOD) environments.

In this section, we compare safe algorithm distillation (AD) (Laskin et al., 2023) and safe algorithm distillation with noise (AD-EPS) (Zisman et al., 2024) in SafeDark-Room environment. AD-EPS aims to learn in-context RL algorithms by training on datasets generated from a single

| Parameter | SafeDarkRoom | SafeDarkMujoco & SafeVelocity |
|---|---|---|
| Embedding Dim | 64 | 512 |
| Hidden Dim | 512 | 256 |
| Num Layers | 8 | 8 |
| Num Heads | 8 | 8 |
| Seq Len | 100 | 200 |
| Attention Dropout | 0.5 | 0.5 |
| Residual Dropout | 0.1 | 0.1 |
| Embedding Dropout | 0.3 | 0.3 |
| Learning Rate | 3e-4 | 3e-4 |
| Betas | (0.9, 0.99) | (0.9, 0.99) |
| Clip Grad | 1.0 | 1.0 |
| Batch Size | 512 | 128 |
| Num Updates | 300k | 500k |
| Optimizer | Adam | Adam |

*Table 4.* Parameters for Safe Algorithm Distillation

optimal policy with injected action noise, enabling efficient collection of learning trajectories.

However, in environments with safety constraints and explicit cost signals, we find that AD-EPS struggles to generalize. In SafeDarkRoom, the perturbed trajectories produced by AD-EPS fail to capture meaningful obstacle-avoidance behavior, as the injected noise primarily induces random action variations rather than structured exploration. As a result, AD-EPS relies on artificial trajectories that do not reflect the true learning dynamics required for effective in-context RL, leading to poor generalization in out-of-distribution settings.

## E. Ablation Studies

In this section, we present our ablation studies on SCARED and safe AD, examining shared factors such as context length and model size, as well as specific factors like dataset size for safe AD. Our findings reveal distinct sensitivities to these factors.

SCARED remains largely unaffected by model size, whereas safe AD performance is significantly influenced by model size (See Figures 5.(b) and 6.(b)). Regarding context length, SCARED benefits from longer sequences, while safe AD shows degraded performance with longer contexts. Conversely, safe AD performs better with shorter context lengths, where SCARED struggles (See Figures 5.(a) and 6.(a)). These results indicate that learning long-term credit assignment is more challenging in offline reinforcement learning due to dataset constraints, whereas online learning, with access to environment interactions, manages long-term credit assignment more effectively. We also confirm the well-established finding that safe AD is highly sensitive to dataset size, with models failing to learn and exhibiting random behavior on out-of-distribution data when the dataset

is small (See Figure 6.(c)).

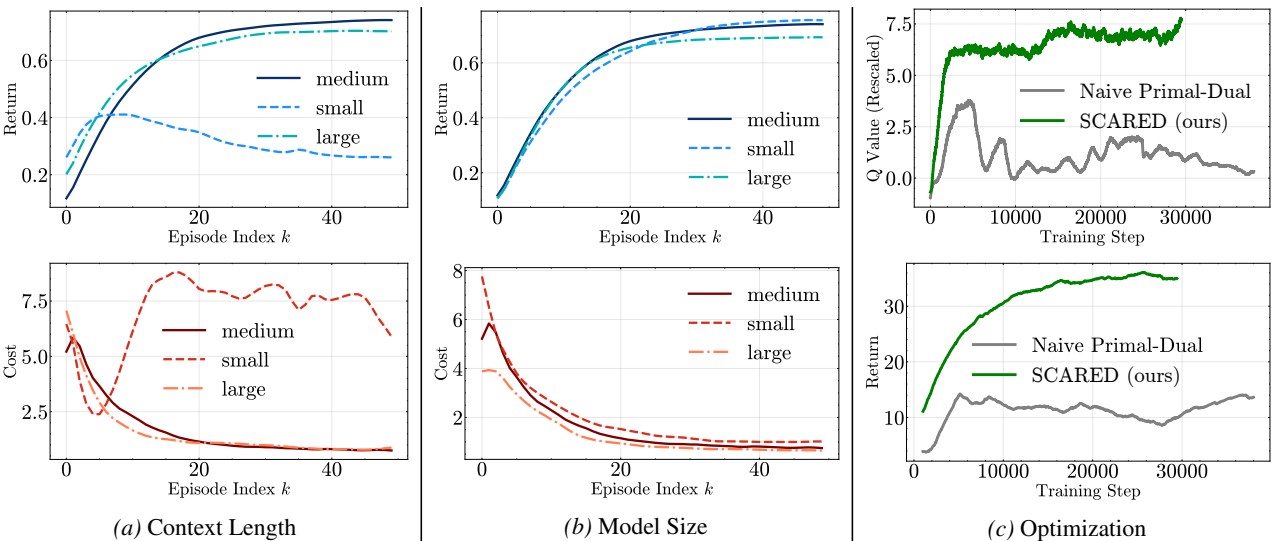

*Figure 5.* **Ablations of SCARED on SafeDarkRoom.** **(a), (b)** The evaluation is set up similarly to Question (i). For context length, we compare 150 and 3000 against the base value of 1500. For model size, we compare embedding dimensions 32 and 128 against the base of 64. **(c)** At each training step, the average total return of 50 episodes across 10 random test environments, and the average Q Value across 50 episodes of 250 random train environments are plotted. Our SCARED optimization is easier to tune and more stable to train than the naive primal-optimal method.

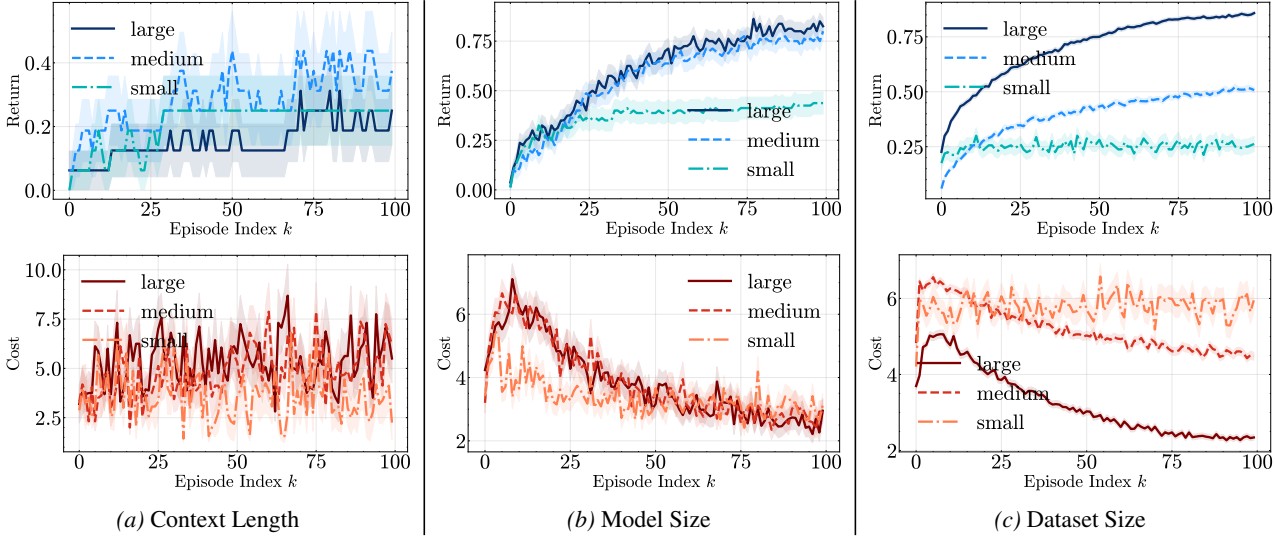

*Figure 6.* **Ablations of Safe Algorithm Distillation on SafeDarkRoom.** The evaluation is set up similarly to Question (i). For context length, we use a smaller base model and test three sequence lengths: 100, 500, and 1000. For dataset size, large refers to the full dataset, medium uses 50% of the dataset, and small uses 5% of the original dataset. Model size increases with the number of hidden layers: 2, 4, and 8, keeping other factors constant.

