# OpenReview forum: "Safe In-Context Reinforcement Learning"
_ICML.cc/2026/Conference — ICML 2026 regular_

### Official Review · Reviewer_ZEhd · 2026-03-10

**Soundness:** 3
**Presentation:** 4
**Significance:** 3
**Originality:** 3
**Overall Recommendation:** 4
**Confidence:** 1

**Summary:**

This paper introduces the first method for Safe In-Context Reinforcement Learning (ICRL). While standard ICRL allows agents to adapt to new environments without test-time parameter updates, previous methods have ignored safety during this learning phase. The authors frame this problem using the Constrained Markov Decision Process (CMDP) framework and propose SCARED (Safe Contextual Adaptive Reinforcement via Exact-penalty Dual). SCARED trains the agent to condition its actions on both its interaction history and its remaining safety budget, allowing it to dynamically maximize rewards while strictly adhering to safety limits during test-time adaptation.

**Compliance With Llm Reviewing Policy:**

Affirmed.

**Final Justification:**

The authors have addressed my questions, and the opinions and suggestions are described above.

**Key Questions For Authors:**

The paper notes that the optimization is stabilized using an exact-penalty dual approach with a single Lagrange multiplier. How sensitive is SCARED's pretraining stability and test-time performance to the initialization or choice of this penalty hyperparameter? Does it require extensive, domain-specific tuning?

Given that online pretraining for sequence models (like Transformers) can be computationally expensive compared to offline algorithm distillation methods, could you provide a detailed comparison of the computational complexity (e.g., wall-clock time, FLOPs) and sample efficiency of SCARED's pretraining versus standard offline ICRL baselines?

**Limitations:**

yes

**Strengths And Weaknesses:**

Significance: The work is highly original in its intersection of two domains: ICRL and Safe RL. The authors are the first to formulate safe ICRL under the CMDP framework. Furthermore, their introduction of challenging OOD benchmarks that require spatial extrapolation (rather than mere interpolation of training distributions) is a valuable contribution that pushes the evaluation standards of the field. The paper is clearly written and logically structured. The background smoothly bridges MDPs, RL, and ICRL, making the problem formulation easy to follow. The distinction between their approach and safe meta-RL (which relies on parameter updates and struggles with context-dependent safety) is clearly articulated.

Weaknesses: While the introduction of extrapolative benchmarks (center-to-edge spawning) is a great step, the environments tested (SafeDarkRoom, for example) are standard tasks. Evaluating the method on tasks with more complex, high-dimensional observation spaces (like pixel-based environments) would further validate the scalability of the exact-penalty dual approach.

The proposed method, SCARED, relies on online reinforcement pretraining. While effective, online pretraining for sequence models can be extremely computationally expensive compared to offline algorithm distillation methods.

---

> ### Author Rebuttal · Authors · 2026-03-31
>
> **Q1. How well would SCARED scale to higher-dimensional observation spaces such as pixel-based environments?**
>
>
> We thank the reviewer for acknowledging the extrapolative benchmarks. We agree that pixel-based observations are a natural next step and plan to explore this in future work. That said, we would like to highlight that SafeDarkMujoco already presents a genuinely challenging setting: the agent's observation does not show obstacle or goal locations, and it must discover them through exploration in a continuous state-action space, relying solely on cost and reward signals, all under a distribution shift.
>
>
>
>
> **Q2. How does SCARED's computational cost compare to offline distillation methods?**
>
>
> We appreciate the question. As reported in Tables 1 and 3 (Appendix C), SCARED is trained for fewer gradient steps than Safe AD. For example, in SafeDarkRoom, SCARED trains for 30k steps (Table 1) compared to Safe AD's 300k steps (Table 3), and on a significantly smaller network (6 million parameters for SCARED vs. 25 million parameters for Safe AD), meaning SCARED consumes far fewer FLOPs overall. Regarding sample efficiency, under full pipeline accounting, Safe AD requires a separate offline data collection stage using safe RL algorithms such as PPO-Lagrangian before distillation, whereas SCARED does not. Thus, when data collection is included, Safe AD uses significantly more environment interaction overall, e.g., 4M environment steps on SafeDarkRoom vs. 1M for SCARED.
>
>
>
>
> **Q3. How sensitive is SCARED to the Lagrange multiplier initialization and tuning?**
>
> We thank the reviewer for this question. In our implementation, the Lagrange multiplier is initialized at zero (Algorithm 1, line 2) and does not require domain-specific tuning. It is then updated adaptively during training: it increases when the cost budget is violated and decreases otherwise (Eq. 6), allowing it to calibrate itself based on the observed constraint violations. This adaptive behavior is one practical advantage of the exact-penalty dual formulation. Theorem 1 indicates that the desired fixed point does not rely on a specially tuned initialization of the multiplier: any primal-optimal policy admits a fixed point with any $\bar{\lambda} \geq\left\|\lambda^*\right\|_{\infty}$. We will clarify this in the revision.

---

> > ### Author Rebuttal · Reviewer_ZEhd · 2026-04-02
> >
> > I have acknowledged the author's response. All questions are answered.

---

> > > ### Author Response · Authors · 2026-04-03
> > >
> > > Dear reviewer ZEhd,
> > >
> > > Thank you for your valuable review. We are pleased to hear that your concerns have been addressed.
> > >
> > > Sincerely, the authors

---

### Official Review · Reviewer_Dehe · 2026-03-11

**Soundness:** 4
**Presentation:** 3
**Significance:** 2
**Originality:** 2
**Overall Recommendation:** 5
**Confidence:** 3

**Summary:**

This paper formalizes safe in-context reinforcement learning (ICRL) under the CMDP framework and proposes SCARED, an online reinforcement pretraining method that satisfies safety constraints at test time without parameter updates. The main technical contribution is a single-multiplier exact-penalty dual formulation that selectively penalizes only constraint-violating episodes. The authors introduce OOD benchmarks for safe in-context adaptation and demonstrate that SCARED outperforms safe algorithm distillation and safe meta-RL baselines across grid-world and continuous control environments.

**Compliance With Llm Reviewing Policy:**

Affirmed.

**Final Justification:**

The rebuttal and follow-up resolved my main concern. I was worried that SCARED's advantage over Safe AD might be confounded by online training and a larger data budget, but the authors clarified that the opposite is true: Safe AD uses substantially more environment steps (~4M vs 1M on SafeDarkRoom) and a much larger model (25M vs 6M parameters). The naive primal-dual ablation in Figure 5c also helps isolate the exact-penalty contribution from online training alone. I am raising my score to accept and ask the authors to make the data and compute budget comparison more explicit in the revised version.

**Key Questions For Authors:**

1. How does SCARED compare to an online version of Safe AD — i.e., a transformer conditioned on RTG and CTG but finetuned with online environment interaction?
2. How should a practitioner select a meaningful CTG target for a genuinely OOD task where the cost structure is unknown?

**Limitations:**

Limitations were not discussed. See weaknesses/questions.

**Strengths And Weaknesses:**

## Strengths

1. The paper clearly formalizes safe ICRL under the CMDP framework, identifying a well-motivated gap: existing ICRL methods do not account for safety constraints. The problem setup is precise and the notation is consistent throughout.

2. The single-multiplier reformulation with [·]₊ masking is a well-motivated fix to the K-multiplier instability problem. The insight that episodes receive uneven update frequencies during pretraining, making per-episode multipliers unstable, is practical and clearly explained.


## Weaknesses

1. SCARED is the only method that uses online environment interaction during pretraining. Safe AD is purely offline, and the meta-RL baselines operate under a different adaptation paradigm. This makes it impossible to determine how much of SCARED's advantage comes from the exact-penalty dual formulation versus simply being an online method. A natural missing baseline is an online decision transformer conditioned on RTG and CTG  (e.g., Safe AD's architecture finetuned with online environment interaction) which would isolate the contribution of the Lagrangian machinery from the benefit of online training.

2. The environments are relatively simple, limiting confidence in how well SCARED would scale to more complex settings.

3. For truly out-of-distribution tasks, it is unclear how a user would specify a meaningful cost budget. If the environment structure is genuinely novel, the relationship between a numerical CTG target and the actual safety behavior may not be well-calibrated. The paper does not discuss this.

---

> ### Author Rebuttal · Authors · 2026-03-31
>
> **Q1. How much of SCARED's advantage comes from the exact-penalty formulation versus simply being an online method?**
>
>
> We thank the reviewer for this thoughtful question. We first note that the meta-RL baselines are also online, and in fact use a stronger adaptation mechanism than SCARED, since they perform parameter updates during adaptation.
>
> More importantly, Figure 5c already provides evidence that SCARED’s advantage is not due to online training alone. The naive primal-dual baseline like [1] is also an online method trained with environment interaction and Lagrangian-based constrained optimization, but it uses a standard per-episode multiplier update instead of SCARED’s exact-penalty dual formulation. Its weaker performance shows that simply training online is not enough; the exact-penalty dual design is a key part of SCARED’s gains.
>
>
>
>
> **Q2. How well would SCARED scale to more complex environments?**
>
>
> We appreciate the reviewer raising this point. While the environments may appear simple at first glance, we believe they should be viewed in the context of prior ICRL and safe meta-RL work. DarkRoom is adopted from the ICRL literature [2, 3, 4] using the hard variant (sparse rewards) with OOD obstacle distributions between training and testing. Velocity is adopted from [5, 6, 7] with speed violation costs similar to [5]. SafeDarkMujoco, which features high-dimensional continuous observations, is adopted from [5] but made substantially harder: the agent's observation does not include obstacle or goal locations, forcing it to infer them solely through exploration in challenging OOD environments (Proposition 1). Thus, we have included both environments that are popular in the literature and benchmarks where we have taken deliberate steps to make them more challenging. SCARED’s strong results across these settings suggest that it is a promising approach beyond only the easiest cases, although we agree that evaluation on more environments of this kind is an important direction for future work.
>
>
>
> **Q3. How should a practitioner select a meaningful CTG for a genuinely OOD task?**
>
> We thank the reviewer for this thoughtful concern. In the CMDP formulation, the cost budget is part of the problem specification. We agree, however, that in a genuinely OOD setting, choosing a meaningful numerical CTG target may not be straightforward in practice. That said, we would like to highlight that when the agent is trained with SCARED, the CTG often approaches zero within each episode as the agent exhausts the given cost budget. As a result, the agent learns very well how to behave conservatively when the CTG is near zero. Consequently, in realistic safety-critical settings, probing CTG starting from zero upward until the desired behavior is reached is a viable approach. As shown in Figure 2, the CTG–Max Cost curve is smooth and well-behaved, supporting this approach.
>
> To illustrate this, we provide a GIF showing 6 consecutive episodes with a very small CTG on an adversarial map, where yellow cells denote obstacles surrounding the agent and the cyan cell denotes the goal. After discovering the dense obstacle structure through a few movements in the first episodes, the agent chooses to remain stationary in the later episodes given the tight cost budget.
> https://osf.io/rs759/files/agwrc?view_only=9810212f4bc5448b8f1caaafd334353d
>
>
>
> ---
>
> [1] Xu et al., CRPO: A New Approach for Safe Reinforcement Learning with Convergence Guarantee, ICML 2021.
>
> [2] Hsu et al., "Sim-to-Lab-to-Real: Safe Reinforcement Learning with Shielding and Generalization Guarantees," Artificial Intelligence, 2023.
>
> [3] Lee et al., "Supervised Pretraining Can Learn In-Context Reinforcement Learning," NeurIPS 2023.
>
> [4] Laskin et al., "In-context Reinforcement Learning with Algorithm Distillation," ICLR 2023.
>
> [5] Xu and Zhu, "Efficient Safe Meta-Reinforcement Learning: Provable Near-Optimality and Anytime Safety," NeurIPS 2025.
>
> [6] Grigsby et al., "AMAGO: Scalable In-Context Reinforcement Learning for Adaptive Agents," ICLR 2024.
>
> [7] Kirsch et al., "Towards General-Purpose In-Context Learning Agents," NeurIPS 2023.

---

> > ### Author Rebuttal · Reviewer_Dehe · 2026-04-01
> >
> > Thank you for the rebuttal. I appreciate the clarification that meta-RL baselines are also online, and the primal-dual baseline in Figure 5c helps partially isolate the exact-penalty contribution. The CTG calibration response is practical and makes sense.
> >
> > However, my core concern from W1 is not fully addressed. Safe AD shares SCARED's transformer-based, CTG-conditioned architecture but is pretrained on offline data only. When SCARED outperforms Safe AD, we cannot tell if the gains come from the exact-penalty dual formulation or from online pretraining. A baseline that would help resolve this is training Safe AD's architecture online.
> >
> > Related to this, can you clarify the total environment steps SCARED uses during pretraining versus Safe AD's dataset size? Since Safe AD is sensitive to dataset size (Figure 6c), a large data budget gap would further confound this comparison.

---

> > > ### Author Response · Authors · 2026-04-03
> > >
> > > We thank the reviewer for the continued discussion. We are glad that our initial response addressed several of the concerns.
> > >
> > > > However, my core concern from W1 is not fully addressed. Safe AD shares SCARED's transformer-based, CTG-conditioned architecture but is pretrained on offline data only. When SCARED outperforms Safe AD, we cannot tell if the gains come from the exact-penalty dual formulation or from online pretraining. A baseline that would help resolve this is training Safe AD's architecture online.
> > >
> > >
> > > We'd like to clarify that Safe AD builds on algorithm distillation [1], which is an inherently offline-based method. Algorithm distillation aims to distill a source algorithm based on its learning trajectories. Thereby, algorithm distillation variants are all offline-based [1, 2, 3, 4, 5]. Moreover, a dataset including learning trajectories of the source algorithm is often collected by on-policy safe RL algorithms, as we did for Safe AD: using PPO-Lagrangian.
> > >
> > > We believe the naive primal-dual ablation in Figure 5c provides the closest comparison the reviewer is looking for: it is a CTG-conditioned architecture, trained with an online method but without the exact-penalty formulation, and its weaker performance demonstrates that the exact-penalty dual design is key to SCARED's gains.
> > >
> > > > Related to this, can you clarify the total environment steps SCARED uses during pretraining versus Safe AD's dataset size? Since Safe AD is sensitive to dataset size (Figure 6c), a large data budget gap would further confound this comparison.
> > >
> > > We thank the reviewer for raising this important point. The comparison is not confounded by SCARED using a larger data budget; in fact, the opposite is true. Under full pipeline accounting, Safe AD uses substantially more compute and more environment interaction than SCARED. For example, on SafeDarkRoom, SCARED trains for 30k updates (Table 1), whereas Safe AD uses 300k updates (table 3). Safe AD also uses a much larger model (25M parameters vs. 6M for SCARED), so its overall compute budget is substantially higher.
> > >
> > > Moreover, Safe AD requires a separate offline dataset collection stage using a safe RL algorithm (e.g., PPO-Lagrangian) before distillation, whereas SCARED does not. Thus, when dataset generation is included, Safe AD also consumes significantly more environment interaction overall. For example, on SafeDarkRoom, the total budget is about 4M environment steps for Safe AD vs. 1M for SCARED. We recognize that this information is currently spread across several tables and sections (Tables 1, 3, and Appendix C), and we will present it clearly in the revised version to better highlight SCARED's efficiency advantage.
> > >
> > > ---
> > >
> > > [1] Laskin et al., "In-context Reinforcement Learning with Algorithm Distillation," ICLR 2023.
> > >
> > > [2] Zisman et al., "Emergence of In-Context Reinforcement Learning from Noise Distillation", ICML 2024.
> > >
> > > [3] Lee et al., "Supervised pretraining can learn in-context reinforcement learning", NIPS 2023.
> > >
> > > [4] Huang et al., "In-Context Decision Transformer: Reinforcement Learning via Hierarchical Chain-of-Thought", ICML 2024.
> > >
> > > [5] Son et al., "Distilling reinforcement learning algorithms for in-context model-based planning", ICLR 2025.

---

### Official Review · Reviewer_5vu4 · 2026-03-13

**Soundness:** 3
**Presentation:** 2
**Significance:** 3
**Originality:** 2
**Overall Recommendation:** 4
**Confidence:** 3

**Summary:**

The paper introduces safe In-Context Reinforcement Learning (ICRL), addressing a critical gap where current models can adapt to new tasks without parameter updates but fail to guarantee safety during exploration. This work uses Constrained Markov Decision Processes (CMDP) to model this problem and proposes a pretraining method named SCARED (Safe Contextual Adaptive Reinforcement via Exact-penalty Dual). Experiments show that SCARED consistently enables safe and robust in-context adaptation, outperforming existing ICRL and safe meta-RL baselines.

**Compliance With Llm Reviewing Policy:**

Affirmed.

**Final Justification:**

All my concerns have been addressed and I have raised my score.

**Key Questions For Authors:**

Please see weaknesses above.

**Limitations:**

I do not find the discussion of the limitations of this work.

**Strengths And Weaknesses:**

Strengths:

- SCARED allows an agent to safely adapt to out-of-distribution tasks relying entirely on its expanding context.

- Themren 1 discusses the maximum separation between training and testing distributions

Weaknesses:

- Lack of the comparision of the context-based meta RL methods like PEARL or Varibad.

- In the proof of Theorem 1, the authors mention that "The full convergence proof to an exact fixed point is beyond the scope of this work." What is the major challenge for proving this result? What is the major insight of Theorem 1?

- Why does the performance of SCARED improve with the increasing of the context length, but the performance of safe AD doesnot imrpove with the increasing of the context length?

- The description of experimental details is unclear. For example, for each domain, what are the training parameters and what are the evaluation parameters? It is better to use a table to clearly describe these parameters of each domain.

---

> ### Author Rebuttal · Authors · 2026-03-31
>
> **Q1. How does SCARED compare to context-based meta-RL methods like PEARL and VariBAD?**
>
>
>
> We appreciate the suggestion. We would like to note that PEARL and VariBAD are unconstrained meta-RL methods that maximize return without any safety consideration, so they solve a fundamentally simpler problem. As shown in Figure 2, SCARED's return naturally increases as the cost budget grows, with unconstrained methods corresponding to the limiting case of an infinite budget. That said, per the reviewer's suggestion, we have evaluated VariBAD on SafeDarkRoom and include the results below. While return improves across episodes, the cost remains high and uncontrolled throughout. This is expected, as VariBAD's algorithm has no mechanism to regulate it, which is precisely the gap that safe ICRL aims to address.
>
> | Episode | Return | Cost |
> |---|---|---|
> | 0 | 0.10 | 9.73 |
> | 5 | 0.23 | 8.75 |
> | 10 | 0.28 | 8.73 |
> | 15 | 0.28 | 9.03 |
> | 20 | 0.35 | 8.08 |
> | 25 | 0.33 | 8.05 |
> | 30 | 0.25 | 9.30 |
> | 40 | 0.33 | 8.55 |
> | 49 | 0.38 | 7.80 |
>
>
>
> **Q2. What is the major insight of Theorem 1, and what is the main challenge for proving convergence?**
>
> We thank the reviewer for this question. The major insight of Theorem 1 is that a single Lagrange multiplier suffices to enforce all K episode-level constraints, making Algorithm 1 both stable and constraint-satisfying.
>
> Regarding the convergence proof, even in standard CMDPs, existing convergence results for primal-dual methods require tabular [1, 2] or linear [3] settings. Our setting introduces additional challenges: the $[g_k(π)]_+$ and $max_k$ operations introduce non-smoothness at the constraint boundary, and unique to ICRL, the constraints $g_1, ..., g_K$ are coupled through the interaction history, creating sequential dependence absent in standard CMDPs. We view a full convergence analysis addressing these challenges as a substantial contribution in its own right, and we leave it as an open question.
>
>
>
> **Q3. Why does SCARED benefit from longer context while Safe AD does not?**
>
> Our hypothesis is that since, unlike SCARED, Safe AD is an imitation-based approach, increasing the sequence length expands the trajectory space that the dataset must cover. Since the sequence length ablation is done by only changing the sequence length while keeping the dataset size fixed, the data coverage drops, leading to degraded performance. This is consistent with our dataset size ablation (Figure 6c), which confirms that Safe AD is highly sensitive to data coverage.
>
>
>
>
>
> **Q4. Could the experimental details be presented more clearly?**
>
> We thank the reviewer for the suggestion. The training and evaluation parameters for each domain are provided in Tables 1–3 (Appendix C) and the experimental setup is described in Sections 5.2–5.3. We agree that putting the training and evaluation distribution parameters for each domain (e.g., obstacle/goal distributions, velocity ranges, CTG ranges) into a single table would improve clarity, and we will add such a table in the revised version.
>
>
>
> ---
>
>   [1] Ding et al., "Natural Policy Gradient Primal-Dual Method for Constrained Markov Decision Processes," NeurIPS 2020.
>
>   [2] Borkar, "An Actor-Critic Algorithm for Constrained Markov Decision Processes," Systems and Control Letters, 2005.
>
>   [3] Ghosh et al., "Provably Efficient Model-Free Constrained RL with Linear Function Approximation," NeurIPS 2022.

---

> > ### Author Rebuttal · Reviewer_5vu4 · 2026-04-03
> >
> > All my concerns have been addressed and I have raised my score.

---

### Official Review · Reviewer_vz9J · 2026-03-17

**Soundness:** 3
**Presentation:** 3
**Significance:** 3
**Originality:** 3
**Overall Recommendation:** 4
**Confidence:** 3

**Summary:**

This paper studies safe in-context reinforcement learning (ICRL), where an agent must adapt to new tasks at test time using only interaction history, without updating its parameters, while also satisfying safety constraints. The authors formulate the problem as a constrained MDP and propose SCARED to promote safe adaptation of ICRL. Extensive experiments and ablations demonstrate the strong performance of SCARED.

**Compliance With Llm Reviewing Policy:**

Affirmed.

**Final Justification:**

I thank the authors for addressing all my concerns. I keep my inital rating.

**Key Questions For Authors:**

* SCARED shows strong performance over several safe-related baselines. However, the paper does not clearly discuss whether these gains may partly stem from differences in the intended application settings or assumptions of the compared methods. Could the authors clarify this point and comment on whether SCARED would remain effective beyond the specific setting considered here?
* The evaluation only considers a bounded range of seemingly reasonable CTG values. The paper does not study infeasible or out-of-support budgets, such as cases where the specified cost budget is incompatible with the task geometry or hazard configuration. So it remains unclear how robust this conditioning interface is under misspecified budgets.

**Limitations:**

Yes

**Strengths And Weaknesses:**

Strengths:
1. This paper builds on parameter-update-free in-context reinforcement learning and identifies the largely unexplored problem of maintaining safety during the trial-and-error adaptation process.
2. Extensive experiments and ablations on several benchmarks validate SCARED's strong performance over other safe-related baselines.
3. The paper provides a relatively rich theoretical analysis, which strengthens the credibility of the proposed method.

Weaknesses:
1. The practical scope of the proposed method appears somewhat narrow. SCARED assumes access to explicit cost signals and a user-specified cost budget (CTG), which may be reasonable in benchmark CMDPs. However, they are harder to define or calibrate in realistic safety-critical applications.
2. The experiments show SCARED's strong performance over other safe-related baselines. However, the comparison of unconstrained ICRL and SCARED-style constrained ICRL is not discussed.

---

> ### Author Rebuttal · Authors · 2026-03-31
>
> **Q1. How practical is it to assume explicit cost signals in realistic safety-critical applications?**
>
> We appreciate the reviewer for raising this point. We follow the standard CMDP formulation adopted in the safe RL literature, where a cost signal is available, just as a reward signal is available in the standard MDP setting. Moreover, we take a step further in our OOD benchmarks by evaluating on cost functions that are out of the distribution the agent has seen during training. In many real-world use cases, some measure of cost, such as pitch/roll limits [1], joint torque/velocity limits [2], or distance-to-obstacle [3], is often available, making our method directly applicable.
>
>  **Q2. How robust is CTG conditioning under misspecified or out-of-support budgets?**
>
>  This is a thoughtful concern. First, we would like to clarify that the cost budget is part of the definition of the CMDP and is expected to be given as part of the problem. While cost signals are readily available, we acknowledge that selecting the right budget for a new task may require calibration. That said, we would like to highlight that when the agent is trained with SCARED, the CTG often approaches zero within each episode as the agent exhausts the given cost budget. As a result, the agent learns very well how to behave conservatively when the CTG is near zero. Consequently, in realistic safety-critical settings, probing CTG starting from zero upward until the desired behavior is reached is a viable approach. As shown in Figure 2, the CTG–Max Cost curve is smooth and well-behaved, supporting this approach.
>
> As illustrated in our response to Reviewer `Dehe` Q3, the agent exhibits sensible conservative behavior under very small CTG budgets.
>
>
> **Q3. How does SCARED compare to unconstrained ICRL, and are there other constrained ICRL methods to compare against?**
>
>
>
>
>
>
>
> We thank the reviewer for this suggestion. We would like to clarify that, to the best of our knowledge, there are no prior constrained ICRL methods. Safe AD, which we include as a baseline, is itself our extension of algorithm distillation to the constrained setting. The safe meta-RL baselines represent the closest available alternatives from adjacent fields. Regarding comparison with unconstrained ICRL, such methods maximize return without any safety consideration and thus solve a fundamentally different problem. As shown in our answer to Reviewer `5vu4` Q1, VariBAD's cost remains uncontrolled, confirming this.
>
>
>
>
>
> **Q4. Could SCARED's gains come from differences in evaluation settings or assumptions across compared methods?**
>
> *Evaluation fairness*
>
> We thank the reviewer for this thoughtful question. All methods are evaluated on well-established environments from the meta-RL and ICRL literature: DarkRoom [3, 4, 5], Velocity [6, 7, 8], and MuJoCo [6, 7], where [6] is the state-of-the-art safe meta-RL work we compare against. While the SafeVelocity environments are adopted from prior work, SafeDarkRoom and SafeDarkMuJoCo under our OOD evaluation present a strictly more challenging setting than prior works (Proposition 1). Beyond the setting considered here, we expect SCARED to remain effective in problems with reward and cost signals availabe at test time, like the standard RL or safe RL setting.
>
> *Method assumptions*
>
> Regarding the compared methods' assumptions, SCARED does not rely on more favorable conditions than the baselines. Unlike the compared methods, it does not require the test-time cost budget to be fixed and known during training; instead, it can adapt to different CTG values at test time (Figure 2). Moreover, the safe meta-RL baselines in [6] use gradient-based updates during adaptation, whereas SCARED is gradient-free, yet still outperforms them. Together, these results suggest that SCARED’s benefits come from its algorithmic design rather than from easier assumptions.
>
>
>
> ---
>
> [1] Yang et al., "Safe Reinforcement Learning for Legged Locomotion," IROS 2022.
>
> [2] Chane-Sane et al., "CaT: Constraints as Terminations for Legged Locomotion Reinforcement Learning," IROS 2024.
>
> [3] Hsu et al., "Sim-to-Lab-to-Real: Safe Reinforcement Learning with Shielding and Generalization Guarantees," Artificial Intelligence, 2023.
>
> [4] Lee et al., "Supervised Pretraining Can Learn In-Context Reinforcement Learning," NeurIPS 2023.
>
> [5] Laskin et al., "In-context Reinforcement Learning with Algorithm Distillation," ICLR 2023.
>
> [6] Xu and Zhu, "Efficient Safe Meta-Reinforcement Learning: Provable Near-Optimality and Anytime Safety," NeurIPS 2025.
>
> [7] Grigsby et al., "AMAGO: Scalable In-Context Reinforcement Learning for Adaptive Agents," ICLR 2024.
>
> [8] Kirsch et al., "Towards General-Purpose In-Context Learning Agents," NeurIPS 2023.

---

> > ### Author Rebuttal · Reviewer_vz9J · 2026-04-03
> >
> > I thank the authors for addressing my concerns. I keep my inital rating.

---

### Decision · Program_Chairs · 2026-04-30

**Decision:**

Accept (regular)

**Comment:**

This paper proposes a CMDP approach to conduct safe ICRL. The review team appreciate both the analytical and empirical insights of the results. I encourage the authors to take the comments into account in the final version.